# Topography based local spherical Voronoi grid refinement on classical and moist shallow-water finite volume models

Luan F. Santos[1] and Pedro S. Peixoto[1]

[1]Instituto de Matemática e Estatística da Universidade de São Paulo, Rua do Matão, 1010 - Butantã, São Paulo - SP, 05508-090

**Correspondence:** Luan F. Santos (luan.santos@usp.br)

**Abstract.** Locally refined grids for global atmospheric models are attractive since they are expected to provide an alternative to solve local phenomena without the requirement of a global high-resolution uniform grid, whose computational cost may be prohibitive. The Spherical Centroidal Voronoi Tesselations (SCVT), as used in the atmospheric Model for Prediction Across Scales (MPAS), allows a flexible way to build and work with local refinement. Alongside, the Andes Range plays a key role in the South American weather, but it is hard to capture its fine structure dynamics in global models. This paper describes how to generate SCVT grids that are locally refined in South America and that also capture the sharp topography of the Andes Range by defining a density function based on topography and smoothing techniques. We investigate the use of the mimetic finite volume scheme employed in the MPAS dynamical core on this grid considering the non-linear classic and moist shallow-water equations on the sphere. We show that the local refinement, even with very smooth transitions from different resolutions, generates spurious numerical inertia-gravity waves that may even numerically de-stabilize the model. In the moist shallow-water model, where physical processes such as precipitation and cloud formation are included, our results show that the local refinement may generate spurious rain that is not observed in uniform resolution SCVT grids. Fortunately, the spurious waves originate from small-scale grid-related numerical errors and therefore can be mitigated using fourth-order hyperdiffusion. We exploit a grid geometry-based hyperdiffusion that is able to stabilize spurious waves and has very little impact on the total energy conservation. We show that, in some cases, the clouds are better represented in a variable resolution grid when compared to a respective uniform resolution grid with the same number of cells, while in other cases, grid effects can affect the cloud and rain representation.

## 1 Introduction

The Andes, the world's longest mountain range, located in entire South America's western coast, acts as an obstacle to the atmospheric circulation and plays a key role in the weather of South America. For instance, the South American low-level jet, formed in the Andes eastern slope, has an important influence in the Brazilian southeast, among other regions, precipitation transporting moisture from the Amazon basin (Garreaud, 2009). Furthermore, the Andes is essential for the formation of the South American low-level jet. (Insel et al., 2010) and Junquas et al. (2015) show that the remotion of the Andes mountain leads to a reduction of the precipitation in the Brazilian southeast. Despite its length, the Andes has a narrow width of less than 200

km almost everywhere in the mountain and its elevation is 4000 m on average but the elevation may suddenly change in a few kilometers.

The Andes complex topography is hard to be captured in a global model due to its abrupt changes in elevation and resolution limitations imposed by computational costs. Furthermore, Figueroa et al. (2016) shows that both Brazilian global Atmospheric Model (BAM) and the Global Forecast System (GFS) produce dry or wet biases in the Amazon basin, over the Andes, and in the Brazilian Southeast. This bias is also observed in other models (Silva et al., 2011; Chou et al., 2020). Therefore, aiming to improve the precipitation forecast in those regions, it is desirable to better represent the Andes Range shape in the employed model. One alternative to overcome the problem of resolution limitations is to employ regional models (e.g. Freitas et al. (2017)). A second alternative is to develop a locally refined global grid that captures well the Andes Range. The second alternative has the advantage of not requiring any artificial boundary conditions as it is usual for regional models and has been developed for some recent global models (Ullrich et al., 2017).

Locally refined grids for global models are built aiming to solve local phenomena without requiring a global high-resolution grid, in which computational cost may be prohibitive. For instance, Barros and Garcia (2004) have proposed a locally refined latitude-longitude grid with a coarse global uniform grid successively refined to a region of interest. Local refinements also have been developed for the cubed-sphere grid. For example, in Harris et al. (2016) a local refinement for the cubed-sphere was developed using stretching transformations, where the region of interest has a resolution 7 times that in the coarse grid region. On other hand, Lean et al. (2008) propose to nest models with resolutions of 12, 4, and 1 km using the UK Met Office operational model at that time, where the 4 and 1 km models use two squares centered in England as the domain. Those works propose feasible ways to have a higher resolution in areas of interest. However, they are usually constrained to a rectangular geometry in the region of interest and impose interpolation constraints between different resolutions, which may affect the dynamics of in/out propagating waves.

Aiming to run at very high resolutions, many recent global atmospheric models are employing quasi-uniform geodesic spherical grids (Staniforth and Thuburn, 2012). The geodesic grids usually have more complex cell geometry, but they can allow more flexibility in local grid refinement. The added geometry complexity on these grids, such as those using Voronoi (hexagonal-pentagonal) cells, makes the development of numerical schemes on these grids more difficult than in regular quadrilateral grids. Nevertheless, the finite volume method for the shallow-water equations proposed by Thuburn et al. (2009) and Ringler et al. (2010), hereafter named TRSK, was designed to work on arbitrary orthogonal Voronoi C-grids. This scheme has many desirable mimetic problems, such as mass and total energy conservation, preservation of stationary geostrophic modes on the f-sphere, among others. It is used in the dynamical core of some atmospheric models such as in the Model for Prediction Across Scale (MPAS) (Skamarock et al., 2012) developed jointly by the National Center for Atmospheric Research (NCAR) and the Los Alamos Laboratory, and the model Icosahedral Dynamical Core (DYNAMICO) (Dubos et al., 2015).

TRSK applies to a variety of grids, in particular, to the Spherical Centroidal Voronoi Tesselation (SCVT) grids which are adopted in MPAS and may be thought of as an optimized version of the hexagonal-pentagonal grid generated from an icosahedral grid. The SCVT grids are orthogonal grids where each grid cell is a Voronoi diagram and the center of mass with respect to a given density function is the corresponding Voronoi region generator (Ju et al., 2011). This grid has the

flexibility of allowing local refinements by carefully defining a density function and employing Lloyd's method (Du et al., 1999, 2003; Ringler et al., 2011). Using a circular refinement region, Park et al. (2014) shows that SCVT Variable Resolution (VR) grids have no significant wave reflection detected in the transition zone between high- and low-resolution regions, which occurs in the nested model Weather Research and Forecasting (WRF) model analyzed in this work. The work of Kramer et al. (2018) investigates three different meteorological events in Europe and concludes that the VR grid used in MPAS global model yields comparable results to the WRF model. Furthermore, the VR grid performed better than the uniform resolution grid in the organized convection test case, being computationally cheaper. VR grids are also feasible in the ocean component of MPAS. For instance, Hoch et al. (2020), considers grids with local refinement in the North American coast and concludes that even when the grid quality is degraded, no spurious effects are observed in the MPAS-Ocean model simulation results. On the other hand, some studies show that the local refinements in the MPAS grid may impact negatively the results of the model. For instance, Rauscher and Ringler (2014) shows that the VR grid generates an eddy kinetic energy maximum in the refined region for the aquaplanet and Held-Suarez test cases. This maximum is caused by the use of the VR grid and is not observed in the uniform resolution grid. Also, in absence of hyperdiffusion, Zhou et al. (2020) reports grid-scale oscillations in a variable resolution grid considering baroclinic wave and tropical cyclones test cases. While MPAS allows a model that is suitable to work with grids with local refinement in arbitrary shapes by only defining a density function, wave representation and interaction in these grids is still not well understood.

In this work, we propose to generate SCVT grids with local refinement based on Andes topography. Our locally refined grids will be designed aiming to capture well the Andes mountain and the South American continent with smooths transitions to the coarse grid regions, focusing on obtaining a grid that may allow better precipitation forecasts in South America. Also, we will perform a detailed analysis of the intrinsic advantages/problems of using TRSK scheme in such grids in two frameworks that capture many important structures of the atmosphere.

In the first framework, we will consider the non-linear shallow-water equations on the sphere and some standard tests proposed in the literature (Williamson et al., 1992; Galewsky et al., 2004) and a more recent test proposed by Shamir et al. (2019). In the second framework, the moist shallow-water model proposed by Zerroukat and Allen (2015) is investigated. This model includes physical processes in the shallow-water model and allows us to simulate rain and clouds. The moist shallow-water model will allow us to study the impact of the local refinement on cloud and rain formation.

This paper is outlined as follows. In Sect. 2, we describe the development of the SCVT grids with local refinement on the Andes mountains. In Sect. 3, the classical and moist shallow-water models are presented, along with their discretization schemes. In Sect. 4 we present the results obtained using TRSK on the developed grids considering the classical and moist shallow-water model. Conclusions are drawn in Sect. 5.

 ## 2 Grids

### 2.1 Spherical Centroidal Voronoi Tesselations

We start by defining the Spherical Voronoi concepts that will be important for the local refinement technique discussed in this work.

Given a set of distinct points on the sphere, that we call generators, $\{\boldsymbol{x_i}\}_{i=1}^{n} \subset \mathcal{S}^2$, we define the i-th Voronoi region as,

95 
$$\Omega_i = \{\boldsymbol{x} \in \mathcal{S}^2 : d(\boldsymbol{x}, \boldsymbol{x_i}) < d(\boldsymbol{x}, \boldsymbol{x_j}), \forall i = 1, \cdots, n, i \neq j\}, \tag{1}$$

where $d(\boldsymbol{x}, \boldsymbol{y})$ is the geodesic distance on the sphere, $\forall \boldsymbol{x}, \boldsymbol{y} \in \mathcal{S}^2$. The sets $\{\Omega\}_{i=1}^{n}$ are called spherical Voronoi tesselation (SVT) and decompose the sphere into disjoint convex spherical polygons and generates a spherical grid (Ju et al., 2011).

With the Voronoi tesselation, we can associate a dual grid by connecting nearest neighbor generators. The geodesic segment connecting neighbors is orthogonal to the sharing edge between these two generators. This orthogonality is important for 100 the numerical schemes that we shall present later. Under the assumption that no 4 near neighbors are co-circular with empty interior, these dual cells all consist of triangles and form a Delaunay triangulation (Okabe et al., 2000). In this duality relation, each Voronoi region corresponds to a unique dual cell vertex, each Voronoi vertex is associated with a unique dual-cell and each Voronoi edge is associated with a unique dual-edge.

An example of SVT is the primal grid of the icosahedral grid, the pentagonal/hexagonal grid. The icosahedral grid is obtained 105 by gnomonically projecting the edges of an icosahedral inscribed within a sphere. This process generates 12 vertices and 20 triangles. Each of these triangles can be divided into 4 new triangles and we can proceed in this manner recursively. Given a level of recursion $l = 0, 1, 2, \cdots$, it can be shown that the icosahedral grid obtained by recursion has $N_l = 10 \cdot 2^{2l} + 2$ vertices, and consequently, the dual pentagonal/hexagonal grid has $N_l$ cells. The number $l$ is also called grid level.

A particular case of SVT, that we are interested in, is the spherical centroidal Voronoi tesselation (SCVT). Given a density 110 function $\rho : \mathcal{S}^2 \rightarrow ]0, \infty[$, for each Voronoi region $\Omega$, the center of mass with respect to $\rho$ is given by:

$$\boldsymbol{x}^* = \frac{\int_\Omega \boldsymbol{x} \rho(\boldsymbol{x}) d\Omega(\boldsymbol{x})}{\int_\Omega \rho(\boldsymbol{x}) d\Omega(\boldsymbol{x})} \tag{2}$$

For this definition of the center of mass, it does not necessarily hold that $\boldsymbol{x}^* \in \mathcal{S}^2$. Therefore, following Du et al. (2003), we introduce the concept of spherical center of mass, which may be calculated using radial projection,

$$\boldsymbol{x}^c = \frac{\boldsymbol{x}^*}{\|\boldsymbol{x}^*\|}. \tag{3}$$

A SCVT is a SVT such that each generator is the center of mass of the corresponding Voronoi region with respect to a given density function $\rho$. As we can notice, SCVT generates spherical grids, that we call SCVT grids. An algorithm that creates SCVT from initial random generators is the iterative Lloyd's method (Du et al., 1999, 2003), which iteratively pushes the Voronoi generators to the Voronoi mass centroid position, and is the method employed in this work.

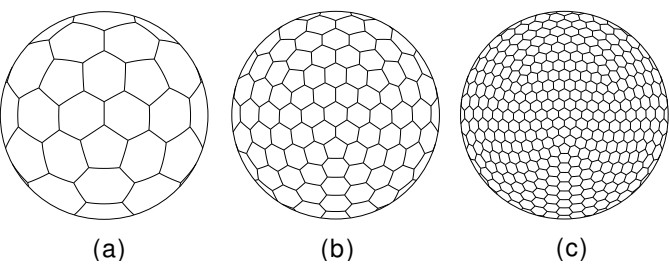

**Figure 1.** SCVT grids using a uniform density function for 12 (a), 42 (b) and 162 (c) generators respectively.

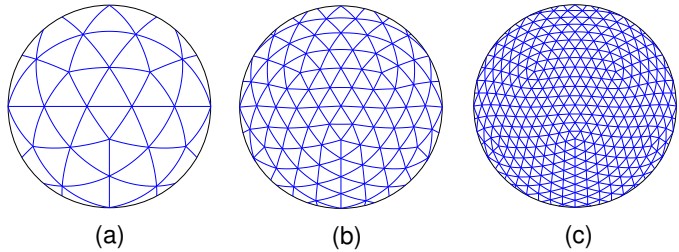

**Figure 2.** Dual (triangular) grids of the grids shown in Fig. 1 for 12 (a), 42 (b) and 162 (c) generators respectively.

Given two Voronoi cells $\Omega_i$ and $\Omega_j$, we denote their diameter by $h_i$ and $h_j$, respectively. It is possible to show that (Du et al., 2003; Ju et al., 2011):

$$\frac{h_i}{h_j} \approx \left( \frac{\rho(\boldsymbol{x_j})}{\rho(\boldsymbol{x_i})} \right)^{\frac{1}{4}}. \tag{4}$$

This estimation plays a key role in this work because it allows us to build grids with local refinement.

We show examples of SCVT's for $\rho = 1$ generated using Lloyd's method in Fig. 1 where we use the pentagonal/hexagonal grid as initial guess in Lloyd's method. The corresponding dual grid is showed in Fig. 2. We use $N_l$ generators, for $l = 2, 3, 4$. The dual grid of SCVT grid may be seen as an optimized version of the icosahedral grid (Miura and Kimoto, 2005).

Next, we present how to build a grid with local refinement on the Andes mountain based in Eq. (4), with the density function presented in Ju et al. (2011) and in the topography data from Amante and Eakins (2009).

## 2.2 Locally refined SCVT

We represent the $\mathbb{R}^3$ coordinates of a point $\boldsymbol{x} \in \mathcal{S}^2$ in spherical coordinates as,

$$\boldsymbol{x} = (\sin \phi \cos \lambda, \sin \phi \sin \lambda, \cos \phi), \quad \phi \in \left[ -\frac{\pi}{2}, \frac{\pi}{2} \right], \lambda \in [-\pi, \pi], \tag{5}$$

and define the parameters $(\phi_c, \lambda_c) = \left( -\frac{\pi}{9}, -\frac{\pi}{3} \right)$, and $\boldsymbol{x_c} = (\sin \phi_c \cos \lambda_c, \sin \phi_c \sin \lambda_c, \cos \phi_c)$. We also consider real values $\gamma$, $\alpha$ and $\varepsilon$ that will be set later. Denoting the Euclidean norm as $\| \cdot \|$, we define the distance from $\boldsymbol{x_c}$ by $d(\boldsymbol{x}, \boldsymbol{x_c}) = \| \boldsymbol{x} - \boldsymbol{x_c} \|$,

for $\boldsymbol{x} \in \mathcal{S}^2$, and introduce the following auxiliary function:

$$
s(\boldsymbol{x}) = \begin{cases} 1 & \text{if } d(\boldsymbol{x}, \boldsymbol{x_c}) \leq \alpha, \\ \frac{\alpha + \varepsilon - d(\boldsymbol{x}, \boldsymbol{x_c})}{\varepsilon} & \text{if } \alpha \leq d(\boldsymbol{x}, \boldsymbol{x_c}) \leq \alpha + \varepsilon, \\ 0 & \text{otherwise.} \end{cases} \tag{6}
$$

A density function that will allow us to refine based on two criteria, in our case the topography (Andes Mountains) and the continent (South America) is given by:

$$
\rho(\boldsymbol{x}) = \frac{1}{\gamma^4} + \left(1 - \frac{1}{\gamma^4}\right)(\mu s(\boldsymbol{x}) + (1 - \mu)b(\boldsymbol{x})), \quad x \in \mathcal{S}^2 \tag{7}
$$

where $b$ is represents the topography (in our case, the Andes region), in this work retrieved from Amante and Eakins (2009) and normalized for the interval $[0, 1]$, $\gamma \geq 1$ and $\mu \in [0, 1]$. Note that for $\mu = 1$, the density function will refine only South America without capturing the Andes mountain sharp topography. For $\mu = 0$, the density function will refine only the Andes mountain sharp topography. Therefore, for a value $\mu \in ]0, 1[$, the generated grid will capture both Andes mountain sharp topography and the South American continent and will have a transition to the coarse global grid. The closer the parameter $\mu$ is to 1, the less the topography will be represented in the refinement. And the closer the parameter $\mu$ is to 0, the more the topography sharp will be represented and less refinement will be in the rest of the South American continent. With $\alpha$, $\mu$ and $s(\boldsymbol{x})$ properly specified, the parameter $\gamma$ will approximately represent the ratio between coarse and fine grid resolution areas, which follows from Eq. (4). We will provide further details at the end of this section.

For the numerical schemes that will be adopted in this worked, sometimes termed as mimetic Finite Volume schemes, grid quality plays a major role. On staggered (C-type, using the meteorology terminology) grids, different variables are placed on different positions of the grid, and operators, differential or not, act on taking variables from one place to another. On highly distorted grids, this can lead to loss of accuracy of some operators (see, for instance, Peixoto and Barros (2013)) or even inadequate physical representation of a model field. One of the key properties of interest is the position of the circumcenters of the Delaunay triangles, which are also the Voronoi cell vertices. Delaunay grids have the so-called *well-centred* property if all circumcenters lie in the interior of their respective triangles (see, for instance, Engwirda (2017)). Here, we will consider grid smoothing techniques to ensure grids with the well-centred property, as we will discuss below.

The ETOPO data (Amante and Eakins, 2009) used in the function $b$ is defined in a latitude-longitude grid with $720 \times 1440$ points, which is enough resolution to represent well the Andes' topography for our experiments. We employed bilinear interpolation to compute the function $b$ in SCVT grid points that are not in the latitude-longitude grid. In order to guarantee grids with smooth transitions between the high and low-resolution regions, we applied a smoothing technique on ETOPO data based on Jacobi method. For each index $(i, j)$ representing a point on the latitude-longitude grid, $0 \leq i \leq 720$, $0 \leq j \leq 1440$, we apply the classic Jacobi iteration process:

$$
b_{ij}^{k+1} = \frac{1}{4}(b_{i-1,j}^k + b_{i+1,j}^k + b_{i,j-1}^k + b_{i,j+1}^k), \tag{8}
$$

where $k = 0, 1, 2, \cdots$ and $b_{ij}^0$ is the initial Andes topography data from ETOPO. The number of iteration was empirically defined as 500 steps, ensuring that the basic shape of the Andes Range is still well represented in the topography dataset. More importantly, this smoothing later ensures that the generated SCVT primal grids have all circumcenters inside the triangles.

We also applied a smoothing filter in the function $s$, considering it defined in the latitude-longitude grid with $720 \times 1440$ points. We used a moving average method by replacing the value of $s$ at a point in the latitude-longitude grid considering the mean of $s$ in a box with $55 \times 55$ points centered at the considered point. This box size showed to guarantee that the generated grids have circumcenters inside the triangles.

We set the parameters to $\gamma = 3$, $\alpha = \frac{7\pi}{45}$, $\varepsilon = \frac{\pi}{12}$ and $\mu = 0.8$. The parameter $\varepsilon$, which represents the width of the circular
refinement region, and the parameter $\gamma$ were defined as in Ju et al. (2011). The parameter $\alpha$ was defined in such a way to guarantee that the circular refinement region contains the South American continent and $\mu$ was chosen empirically aiming to represent well the Andes shape. In Fig. 3, we show the grid generated with 10242 nodes after applying Lloyd's method. The diameters of cells are also shown in Fig. 3, including the diameters of the grid generated with 163842 nodes. The diameter of a Voronoi cell is estimated as the average distance from its generator to its neighbors. As we can notice, the ratio of the diameters
of the cells on the Andes and the cells in the coarser grid region is approximately 3. We can derive this from Eq.(4) and Eq.(7). Indeed, observe that if $x_i \in \mathcal{S}^2$ is a Voronoi generator such that $d(x_c, x_i) > \alpha + \varepsilon$ , i.e., $x_i$ is in the uniform resolution region, we have $s(x_i) = b(x_i) = 0$ and therefore $\rho(x_i) = \frac{1}{\gamma^4}$. On the other hand, notice that if $x_j \in \mathcal{S}^2$ is a Voronoi generator and a point in the Andes region, we have $s(x_j) = 1$ and $b(x_j) \approx 1$, therefore $\rho(x_j) \approx 1$, using Eq.(7). At last, using Eq.(4), we can see that $\frac{h_i}{h_j} \approx \frac{1}{\gamma}$, where $h_i$ and $h_j$ are the diameters of the Voronoi regions $i$ and $j$, respectively.
We generate the variable resolution SCVT grids using the density function defined in Eq. (7) using $N_l = 10 \cdot 2^{2l} + 2$ generators, for $l = 1, 2, \cdots, 7$ and we refer to these grids as VR1, VR2, $\cdots$, VR7, respectively. These generated grids and the uniform SCVT grids will be used in our numerical experiments in Sect. 4. The uniform resolution SCVT grids were generated from grid level 1 up to 9 and we refer to these grids as UR1, UR2, $\cdots$, UR9. We show the minimum and maximum diameters of the cells for the uniform and variable resolution SCVT grids in Table 1. We also generated locally refined grids with a 1:6 ratio
of diameters between coarse grid and cells at the Andes region, with $\gamma = 5$, $\alpha = \frac{7\pi}{45}$, $\varepsilon = \frac{\pi}{12}$ and $\mu = 0.8$. These grids will be analyzed in our numerical experiments only in the Sect. 4.2.1 aiming to investigate the impact of increasing the resolution of the refined region.

# 3   Model description

## 3.1   Shallow-water model

As it is usual in the development of numerical methods for global atmospheric models, we first evaluate it considering the shallow-water equations on the sphere. These equations represent important aspects of the atmosphere, such as Rossby waves, Coriolis effect and geostrophic adjustment, and they have the advantage of being two-dimensional, therefore computationally cheaper to run at high-resolutions.

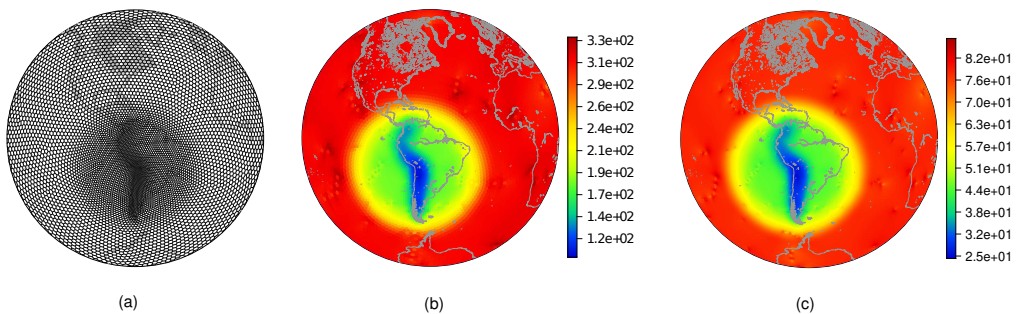

**Figure 3.** SCVT grid with local refinement on Andes mountain using 10242 generators (a). Cell diameters in kilometers for the grid with (b) 10242 and (c) 163842 generators considering $\gamma = 3$, $\alpha = \frac{7\pi}{45}$, $\varepsilon = \frac{\pi}{12}$ and $\mu = 0.8$.

| Grid level | Number of Voronoi cells | Voronoi cells diameters (km) of SCVT grids | | | | | |
| --- | --- | --- | --- | --- | --- | --- | --- |
| | | Uniform resolution | | | Variable resolution | | |
| | | Grid name | Min | Max | Grid name | Min | Max |
| 1 | 42 | UR1 | 4084 | 4649 | VR1 | 2479 | 5189 |
| 2 | 162 | UR2 | 2035 | 2283 | VR2 | 858 | 2672 |
| 3 | 642 | UR3 | 996 | 1128 | VR3 | 425 | 1320 |
| 4 | 2562 | UR4 | 484 | 562 | VR4 | 198 | 673 |
| 5 | 10242 | UR5 | 235 | 281 | VR5 | 98 | 336 |
| 6 | 40962 | UR6 | 114 | 140 | VR6 | 47 | 172 |
| 7 | 163842 | UR7 | 55 | 70 | VR7 | 24 | 87 |
| 8 | 655362 | UR8 | 27 | 35 | - | - | - |
| 9 | 2621442 | UR9 | 14 | 17 | - | - | - |

**Table 1.** Minimum and maximum diameters of the Voronoi cells for the uniform SCVT grid (UR) and for the variable resolution (VR) grid that captures well the Andes mountain. The VR grids considers the parameters $\gamma = 3$, $\alpha = \frac{7\pi}{45}$, $\varepsilon = \frac{\pi}{12}$ and $\mu = 0.8$.

We denote a point on the sphere by $\boldsymbol{x}$ and a time instant by $t \geq 0$. Then, we consider the non-linear shallow-water equations on the sphere written in the vector invariant form (Ringler et al., 2010):

$$\frac{\partial \boldsymbol{u}}{\partial t} = -qh\boldsymbol{u}^{\perp} - \nabla B, \tag{9}$$

$$\frac{\partial h}{\partial t} = -\nabla \cdot (h\boldsymbol{u}), \tag{10}$$

where $h = h(\boldsymbol{x}, t)$ denotes the total fluid depth, $\boldsymbol{u} = \boldsymbol{u}(\boldsymbol{x}, t)$ is the velocity vector tangent to the sphere for all $t \geq 0$, and all other terms are defined in Table 2. The operators $\nabla$ and $\nabla \cdot$ are assumed to be the horizontal gradient and divergence on the sphere, respectively. These equations define an initial value problem that describes the evolution of a thin layer of a two-dimensional fluid on the sphere.

| Variable description | Simbol | Definition |
|---|---|---|
| Bottom topography | $b$ | - |
| Earth gravity | $g$ | $9.8 \text{ m s}^{-2}$ |
| Earth rotation velocity | $\Omega$ | $7.292 \times 10^{-5} \text{ rad s}^{-1}$ |
| Latitude | $\phi$ | - |
| Local unit vector pointing in vertical such that $\boldsymbol{u} \cdot \boldsymbol{k} = 0$ | $\boldsymbol{k}$ | - |
| Perpendicular velocity vector | $\boldsymbol{u}^{\perp}$ | $\boldsymbol{k} \times \boldsymbol{u}$ |
| Coriolis parameter | $f$ | $2\Omega \sin\phi$ |
| Kinetic energy | $K$ | $\|\boldsymbol{u}\|^2/2$ |
| Bernoulli potential | $B$ | $g(h+b) + K$ |
| Relative Vorticity | $\zeta$ | $\boldsymbol{k} \cdot \nabla \times \boldsymbol{u}$ |
| Absolute vorticity | $\eta$ | $\zeta + f$ |
| Potential vorticity | $q$ | $\frac{\eta}{h}$ |

**Table 2.** Variables and parameters for the shallow-water model

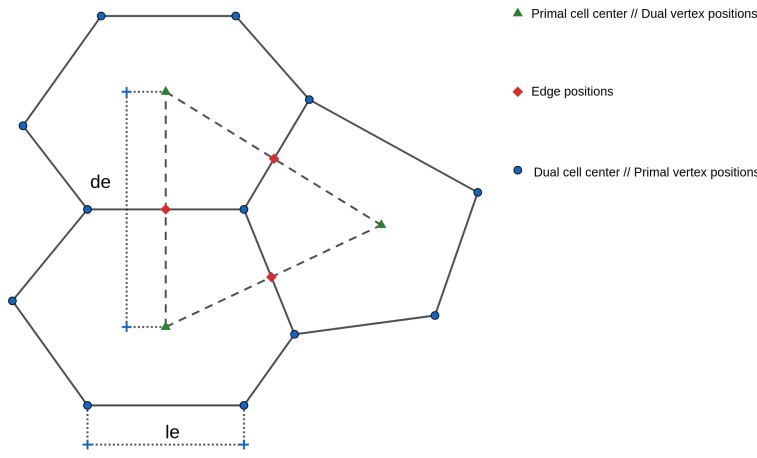

**Figure 4.** Variable positions for a C-grid staggering.

The finite volume scheme adopted here was proposed in Thuburn et al. (2009) and Ringler et al. (2010), also denoted as TRSK, and considers a grid with an orthogonal dual grid and a C-grid staggering of the variables. We consider three positions in the C-grid: primal cell centers (or dual-cell vertices), primal cell vertices (or dual-cell centers), and the intersection point between a primal edge and dual-edge that we call edge points (Fig. 4).

We summarize the notation needed for TRSK in the Table 3. In TRSK, the height $h$ is stored at primal centers and is denoted by $h_i(t) = h(\boldsymbol{x_i}, t)$. Given an edge $e$, we define a normal vector $\boldsymbol{n}_e$ to the edge pointing to the direction such that the normal

| Description | Simbol |
| --- | --- |
| Primal cell | $i$ |
| Dual cell | $v$ |
| Primal/dual edge | $e$ |
| Center of a primal cell $i$ | $\boldsymbol{x_i}$ |
| Edge point of an edge $e$ | $\boldsymbol{x_e}$ |
| Center of a dual cell $v$ | $\boldsymbol{x_v}$ |
| Primal edge length | $l_e$ |
| Dual edge length | $d_e$ |

**Table 3.** Grid notation following Ringler et al. (2010).

component of the velocity given by $u_e(t) = \boldsymbol{u}(\boldsymbol{x_e}, t) \cdot \boldsymbol{n}_e$ is positive. The normal component of the velocity is stored at edge points. Our prognostic variables in TRSK are $h_i$ and $u_e$ for each primal center $i$ and edge $e$. Taking the dot product of Eq. (9) with $\boldsymbol{n}_e$ and get:

$$\frac{\partial u_e}{\partial t} = -[qh\boldsymbol{u}^\perp]_e - [\nabla B]_e, \tag{11}$$

$$\frac{\partial h_i}{\partial t} = -[\nabla \cdot (h\boldsymbol{u})]_i, \tag{12}$$

where $[\boldsymbol{F}]_e = \boldsymbol{F}(\boldsymbol{x_e}, t) \cdot \boldsymbol{n_e}$ and $[G]_i = G(\boldsymbol{x_i}, t)$, for any vector field $\boldsymbol{F}$ and scalar field $G$. Each term on the right hand side of Eq. (11) and Eq. (12) are approximated using the spatial discrete operators from Ringler et al. (2010). After discretizing each term for all primal cells and edges, we get an ordinary differential equation system. In this work, we employed a fourth-order Runge-Kutta scheme to solve the ordinary differential equation system since we are aiming to analyze only the impact of the local refinement on the horizontal discretization. At last, TRSK has been known for its mimetic properties, i.e, the discrete equations mimics some of the continuous equations, such as:

1. Mass conservation;

2. Coriolis term is not an energy source/sink $(u_e u_e^\perp = 0)$;

3. Total energy is conserved within time error truncation;

4. Preservation of stationary geostrophic modes on the $f$-sphere.

The TRSK discretization is also known to be very low order accurate, and may even lack consistency on nonregular grids (Peixoto and Barros, 2013; Peixoto, 2016). However, due to its mimetic properties and flexibility, it is still adopted in some modern global atmospheric models such as MPAS (Skamarock et al., 2012) and DYNAMICO (Dubos et al., 2015).

## 3.2 Moist shallow-water model

Even though the shallow-water model retains keys properties of the atmosphere, they lack the representation of sub-scale physical processes. Therefore, we consider here an intermediate step of a two-dimensional moist shallow water model, avoiding the computational dispendure of a three-dimensional primitive equation model while being able to account for water-related sub-grid processes.

We follow the moist shallow-water model proposed by Zerroukat and Allen (2015), where moist dynamics are included in the classic shallow water-model. This model is two-dimensional, which is computationally cheaper than three-dimensional models at high resolutions, and is derived from the Boussinesq approximation of the three-dimensional Euler equations. The model can be summarized in the following equations:

$$\frac{\partial \boldsymbol{u}}{\partial t} = -qh\boldsymbol{u}^{\perp} - \nabla B + S_u, \tag{13}$$

$$\frac{\partial h}{\partial t} = -\nabla \cdot (h\boldsymbol{u}), \tag{14}$$

$$\frac{\partial h\theta}{\partial t} = -\nabla \cdot (h\theta\boldsymbol{u}) + hS_\theta, \tag{15}$$

$$\frac{\partial hq^k}{\partial t} = -\nabla \cdot (hq^k\boldsymbol{u}) + hS_q^k \tag{16}$$

where $\theta$ is the temperature, $k = 1, 2, 3$, $q^1 = q_v$ is the water vapor state, $q^2 = q_c$ is the cloud state, $q^3 = q_r$ is the rain state. We are using the flux form for the advection equations for numerical purposes that we shall see later. The moisture variables are tracers that are advected in this system. The other variables are the same as in the shallow-water model (Table 2). The source terms of the advection equations define a three-state moist physics, where vapor may be converted to clouds, clouds are allowed to evaporate and clouds may be converted to rain. This three-state moist physics depends on a saturation function, which is given by Zerroukat and Allen (2015):

$$q_{sat}(\theta) = \frac{q_0}{g(h+b)} \exp(20\theta), \tag{17}$$

and the initial vapor state is given by $q_v(\lambda, \phi, 0) = G(\lambda, \phi)q_{sat}(\theta)$, where $\lambda$ denotes the longitude and $q_0$ is a constant chosen in order to guarantee that the initial maximum value of $q_v$ is 0.02 and $G$ is a function between 0 and 1. The additional variables and parameters to the moist shallow water model are shown in Table 4. Note that $S_{q_v} + S_{q_c} + S_{q_r} = 0$, therefore the integral of $h(q_v + q_c + q_r)$ is conserved in this model.

| Variable description | Symbol | Definition |
|---|---|---|
| Precipitation threshold | $q_{precip}$ | $10^{-4}$ |
| Pseudo latent heat constant | $L$ | $10$ |
| Vapor-cloud conversion rate | $\gamma_v$ | $(1 + L\frac{\partial q_{sat}}{\partial \theta})^{-1}$ |
| Cloud-rain conversion rate | $\gamma_r$ | $10^{-3}$ |
| Time step used in the numerical model | $\Delta t$ | - |
| Vapor-cloud conversion | $\Delta q_v$ | $\max\{0, \gamma_v(q_v - q_{sat})\}/\Delta t$ |
| Cloud evaporation | $\Delta q_c$ | $\min\{q_c, \max\{0, \gamma_v(q_v - q_{sat})\}\}/\Delta t$ |
| Cloud-rain conversion | $\Delta q_r$ | $\max\{(0, \gamma_r(q_c - q_{precip})\}/\Delta t$ |
| - | $\Pi$ | $\frac{1}{2}h^2\theta$ |
| Momentum equation source | $S_u$ | $g\theta\nabla b + \frac{1}{h}\nabla\Pi$ |
| Temperature equation source | $S_\theta$ | $L(\Delta q_v - \Delta q_c)$ |
| Vapor equation source | $S_q^v$ | $\Delta q_c - \Delta q_v$ |
| Cloud equation source | $S_q^c$ | $\Delta q_v - \Delta q_c - \Delta q_r$ |
| Rain equation source | $S_q^r$ | $\Delta q_r$ |

**Table 4.** Variables and parameters for the moist shallow-water model (Zerroukat and Allen, 2015)

In this model, we consider the prognostic variables are $\boldsymbol{u}$, $h$, $h\theta$ and $hq^k$, $k = 1, 2, 3$. Similarly to the shallow water model, we store $h$, $h\theta$ and $hq^k$ at Voronoi centers and the normal component of $\boldsymbol{u}$ is stored at the edges. We write similar equations to Eq. (11) and Eq. (12) for the moist shallow-water equations and discretize the right hand side using TRSK operators.

The divergence term in Eq. (15) and Eq. (16) for $h\theta$ and $hq^k$ may be computed using the divergence discretization of TRSK, and this explains why we used the advection equation in flux form. The source terms for Eq. (15)-(16) are straightforward to 255    calculate once they depend only on $\theta$ and $q^k$, thus we only need to divide these variables by $h$. Equation (13) has a source term and it needs to be evaluated at edge points. This source may be calculated using TRSK interpolation and gradient operators. We point out that MPAS solves the tracer equations such as Eq. (15)-(16) using a high-order advection scheme (Skamarock and Gassmann, 2011). Here, we consider the TRSK scheme to solve tracer equations aiming to analyze how TRSK simulates the formation of cloud and rain. Furthermore, as we shall see in Sec. 4.3, a monotonic (positivity preserving) filter is employed 260    in our simulations.

## 4    Numerical Results

This section is dedicated to present the results of the TRSK method on the variable resolution grids considering two frameworks: the classical shallow-water model and the moist shallow-water model developed by Zerroukat and Allen (2015). We will consider the variable resolution grids developed in Sect. 2.2 and the uniform SCVT grids to compare the impact of introducing 265    a local refinement. We also consider the addition of variable numerical fourth-order hyperdiffusion on the locally refined grids

tests. It will be clear whether we are adding hyperdiffusion or not. We give details about the hyperdiffusion employed in the numerical experiments in Sect. 4.1, and we show the results for the shallow-water model and for the moist shallow-water model in Sections 4.2 and 4.3, respectively. The $L_2$ error will be refereed as root mean square error (RMS, hereafter) and the $L_\infty$ error as the maximum error.

## 4.1 Hyperdiffusion

As we will show in the next sections, the variable resolution grid triggers unstable modes of the numerical method and spurious waves are generated, destabilizing the numerical solution. While undesirable for the shallow-water equations, artificial numerical diffusion can be used to mitigate the problem. Here we discuss the numerical diffusion mechanisms that will be explored in this study. To ensure selectiveness of the diffusion mechanism, we will employ 4th order diffusion (hyperdiffusion), but results with second order difusion may be found in an earlier version of the work, available from the journal open discussion material.

The numerical hyperdiffusion was included in the momentum equation (Eq. (11)), following Skamarock et al. (2012) and Jablonowski and Williamson (2011), based on the following vector identity for the vector Laplacian,

$$\Delta \boldsymbol{u} = \nabla(\nabla \cdot \boldsymbol{u}) - \nabla \times (\nabla \times \boldsymbol{u}). \tag{18}$$

Evaluating the above identity on an edge $e$ and taking the dot product with corresponding normal vector $\boldsymbol{n}_e$, we have,

$$[\Delta \boldsymbol{u}]_e = [\nabla(\nabla \cdot \boldsymbol{u})]_e - [\nabla \times (\nabla \times \boldsymbol{u})]_e. \tag{19}$$

The first term on the right hand side of Eq. (19) is calculated using the gradient at edges in normal direction using the divergence given at Voronoi centers. The second term of the right hand side of Eq. (19) is calculated as $\nabla \zeta(\boldsymbol{x}_e) \cdot \boldsymbol{t}_e$ (Skamarock et al., 2012), where $\boldsymbol{t}_e = \boldsymbol{k} \times \boldsymbol{n}_e$, and is obtained as the gradient at edges in tangent direction using the relative vorticity $\zeta$ given at primal vertex positions. The fourth-order hyperdiffusion $\nabla^4$ is obtained using Eq. (19) and the relations $\nabla^2 = \Delta$ and $\nabla^4 \boldsymbol{u} = \nabla^2(\nabla^2 \boldsymbol{u}) = \Delta(\Delta \boldsymbol{u})$.

In order to introduce a variable coefficient of hyperdiffusion, we follow Klemp (2017) using the formula:

$$(\nabla \cdot K_2 \nabla)\boldsymbol{u} \rightarrow \nabla(K_2 \nabla \cdot \boldsymbol{u}) - \nabla \times (K_2 \nabla \times \boldsymbol{u}). \tag{20}$$

where $K_2$ denotes the variable diffusion coefficient. This formula (20) is not exact. However, as Klemp (2017) shows, this formulation ensures a dissipative behavior. This formulation allows us to discretize the Eq. (20) just as we discretized Eq. (19). Given a variable hyperdiffusion coefficient $K_4$, we may obtain a fourth-order filter using Eq. (20) recursively with $K_2 = \sqrt{K_4}$, as explained by Klemp (2017). After the hyperdiffusion in computed, we add it on the right-hand side of momentum equation (Eq. (11)) considering a negative sign to ensure the dissipative behavior the hyperdiffusion operator.

We have examined different possibilities of hyperdiffusion mechanisms. First, we considered constant and cell size (diameter-based) hyperdiffusion coefficients in Sect. 4.2.1. The diameter-based hyperdiffusion coefficient ($K_4$) used in this work follows Zarzycki et al. (2014) and, in a Voronoi cell $i$, is given by $K_{\max} \cdot \left(\frac{h_i}{h_{\max}}\right)^p$, where $h_i$ is the diameter of the $i-$th Voronoi cell,

$h_{\max}$ is the maximum diameter of all cells and $p = \log_2 10$. With this choice of $p$, the hyperdiffusion coefficient reduces 10 times when the cell diameter is divided by 2.

As will be shown in the numerical results, the dominant factor for larger errors in the numerical solution is the cell geometry. With the variable resolution grid, the occurrence of more distorted cells is almost inevitable, and these are also responsible for triggering the numerical instabilities. Peixoto and Barros (2013) showed that one can detect such problematic cells following an "alignment index", which measures how well the cell edges are aligned. Perfectly aligned cells have edges 2-by-2 parallel and of the same length, ensuring second-order accuracy of typical finite volume operators. Peixoto (2016) shows how the grid alignment can affect the solution of the shallow-water equations, and Peixoto and Barros (2014) used the alignment index to build vector reconstructions that use different schemes depending on the cell geometry. Following similar reasoning, here we propose a variable hyperdiffusion that adapts the coefficient based on this alignment index. We, therefore, employ less diffusion in better-aligned cells, so we can be more selective on imposing the diffusion mechanism where it is really needed.

For the alignment-based hyperdiffusion, we considered a variable coefficient $K_4$ which, in a Voronoi cell, is given by a smoothed alignment index (normalized between 0 and 1) multiplied by a value $K_{\max}$ for each cell. The alignment index is defined in Peixoto and Barros (2013) and the value of 0 indicates a perfectly aligened cell, therefore, on almost perfectly aligned cells the diffusion coefficient will be close to zero. The value $K_{\max}$ denotes the maximum of fourth-order hyperdiffusion coefficient allowed is is employed in the most ill-aligned cells. To avoid abrupt transitions in the diffusion mechanism, here the alignment index is smoothed by replacing its cell value by the average of its neighbors (including its own value). The smoothed alignment on edges and vertex positions may be obtained using TRSK interpolation operators.

The values of $K_{\max}$ used in this work were chosen after an investigation of the geostrophically balanced flow test case from Williamson et al. (1992), looking for the minimum coefficient that could minimize the error of the normal component of the velocity. We describe this experiment in details in Sect. 4.2.1.

We highlight that other, different, diffusion mechanisms were investigated in this work, but not shown here. Second-order Laplace diffusion was applied in the early stages of this work, with adequate stabilization of the model, however, with greater impacts on accuracy and energy conservation. Vorticity base stabilization mechanisms (Weller, 2012) were also explored, but these generally failed to stabilize the model (see specific comments in Sect. 4.2.3).

## 4.2 Shallow-water model

### 4.2.1 Steady geostrophic flow

This test case is proposed in Williamson et al. (1992) and it is known as test case 2 (hereafter, TC2). The initial conditions are given by:

$$h(\lambda, \phi, 0) = h_0 - \frac{1}{g}\left( a\Omega u_0 + \frac{u_0^2}{2} \right)\sin^2\phi, \tag{21}$$

$$u(\lambda, \phi, 0) = u_0 \cos\phi, \tag{22}$$

$$v(\lambda, \phi, 0) = 0. \tag{23}$$

The bottom topography is equal to zero. The functions defined above satisfy the shallow water equations and represent a balanced geostrophic flow. Thus, the initial condition should remain constant. We define the parameters $h_0 = 3 \times 10^3$ and
$u_0 = \frac{2\pi a}{12 days}$, where $a$ is the Earth radius.

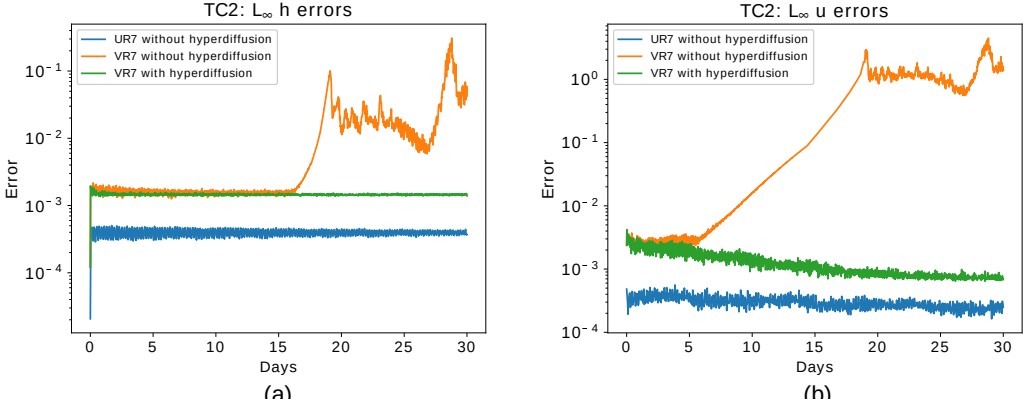

**Figure 5.** Steady geostrophic flow test case: Time evolution of relative error in the $L_\infty$ norm for the height field (a) and the normal component of the velocity field (b). The blue lines indicate the errors for the UR7 grid, while the orange lines show the error for the VR7 grid with refinement based on topography and the green lines show the error for the same VR7 grid but adding numerical fourth-order hyperdiffusion based on the alignment index with $K_{\max} = 10^{12} m^4 s^{-1}$ in the shallow-water model.

In Fig. 5 we show the evolution of the relative error in norm $L_\infty$ for the grids of level 7 (UR7 and VR7). This result shows that the error for the normal component of the velocity field grows since early stages of the integration for the VR7 grid without numerical hyperdiffusion. This error triggers errors in the height field that start to grow significantly after day 15. This behavior does not occur in the UR7 grid. A similar error evolution shown in Fig. 5 may be obtained for the VR6 grid (not shown). There 335  is, therefore, a clear need of artificial numerical diffusion mechanisms for stabilization purposes.

To decide on the best hyperdiffusion strategy, we now examine the different formulations: constant-coefficient, a diameter-based variable coefficient, and an alignment-based variable coefficient. For the variable resolution grid of level 6 (VR6), we show in Fig. 6 the height field error after 30 days considering constant and diameter-based hyperdiffusion coefficients considering $K_{\max} = 10^{13} m^4 s^{-1}$. The error pattern of both solutions (Fig. 6 a and b) have a clear correspondence with the alignment 340  index (Fig. 6 c). However, we note that the error patterns are not necessarily related to the grid resolution, indicating that, while diameter-based hyperdiffusion may be well justified from a physical point of view, it may not be ideal for stabilization requirements originated from grid irregularities.

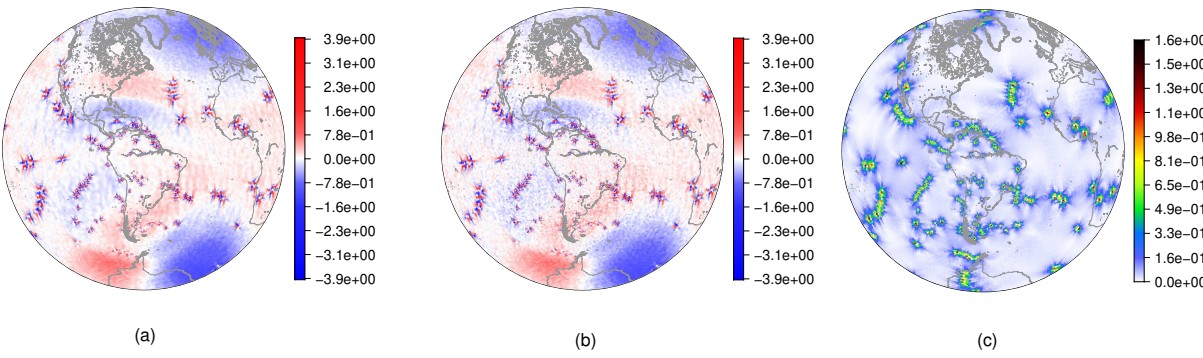

**Figure 6.** (a) Height field error using constant hyperdiffusion coefficient and (b) diameter-based hyperdiffusion coefficient for the steady geostrophic flow test case considering a VR6 grid at day 30 and $K_{\max} = 10^{13} m^4 s^{-1}$ (see Sect. 4.1). (c) Alignment index (Peixoto and Barros, 2013) for VR6 grid.

In Fig. 7 we present the relative errors after 30 days for the height and velocity fields in $L_\infty$ and $L_2$ norms considering the VR6 grid, with respect to different hyperdiffusion coefficient choices ($K_{\max}$) for the constant, diameter-based and alignment-based hyperdiffusion described in Sect. 4.1. We can notice from this figure that the error of velocity field requires more hyperdiffusion to be stabilized, for all the employed methods. Also, the error of diameter-based coefficient requires higher values of $K_{\max}$ to stabilize the error of the height field. The stabilization behavior of the alignment-based method is more similar to the behavior of the constant coefficient, while imposing diffusion only where necessary. This motivated us to move forward with the alignment-based hyperviscosity for further experiments.

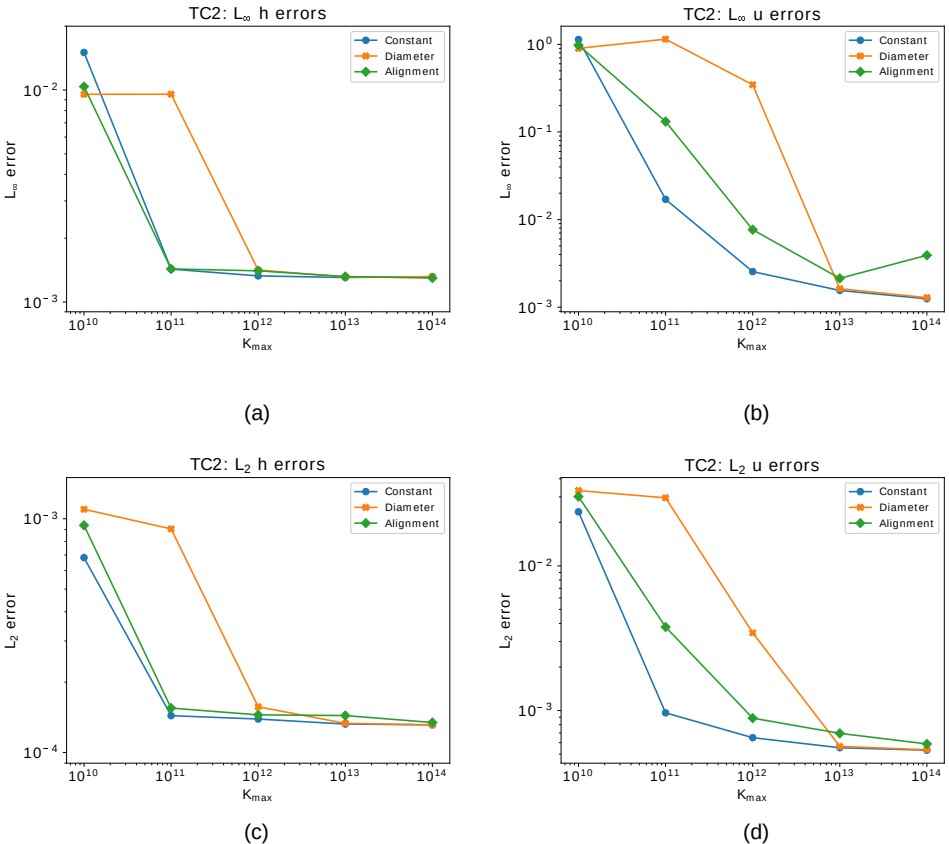

**Figure 7.** Steady geostrophic flow test case showing relative errors after 30 days for the VR6 grid using the different hyperdiffusion methods and different values of $K_{\max}$. (a) $L_\infty$ error for the height field. (b) $L_\infty$ error for the normal component of the velocity field. (c) $L_2$ error for the height field. (d) $L_2$ error for the normal component of the velocity field. The blue lines consider constant hyperdiffusion coefficient, the orange lines consider alignment-based coefficient, and green lines consider diameter-based coefficient.

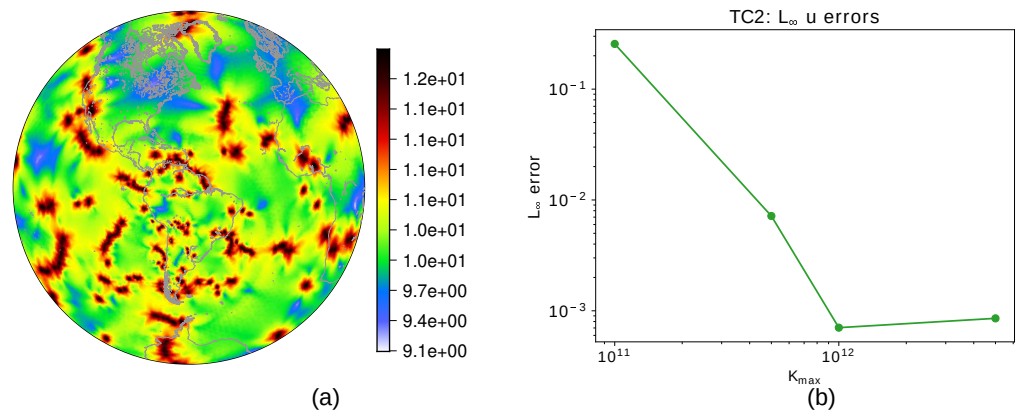

(a)    (b)

**Figure 8.** Hyperdiffusion coefficient (in $\log 10$ scale) based on the smoothed alignment index for each cell in the VR7 grid (a) and $L_\infty$ error of the normal component of the velocity field for different values of $K_{\max}$ considering the steady geostrophic flow test case for the VR7 grid (b).

For a variable resolution grid of level 7 (VR7), an optimal choice for hyperdiffusion coefficient may be found at around $K_{\max} = 10^{12} m^4 s^{-1}$, as shown in Fig. 8b, where we also show the distribution of the coefficient on the sphere (Fig. 8a). Note that large areas of the globe are dominated by coefficients with orders of magnitude smaller than $K_{\max}$ (ranging around $10^9$ to $10^{10} m^4 s^{-1}$). With this approach, we see from in Fig. 5 how the model is now stable in the period of 30 days, with the error evolution in the VR7 grid with numerical hyperdiffusion having similar behavior to the UR7 grid.

Fig. 9 shows the error after 30 days for the height field in the VR7 grid without (Fig. 9a) and with alignment-based hyperdiffusion (Fig. 9b). We also show the alignment index (Fig. 9c). From Fig. 9a, we can see that the error for the height field propagates similarly to gravity waves, although spurious, when no numerical fourth-order hyperdiffusion is employed. From Fig. 9b and 9c, it is clear that the error obtained for the height field is greater in the cells that are ill aligned, and the same holds for the UR7 grid, as Fig. 9d-e show. This result is in agreement with the results from Peixoto (2016), where they show that the

divergence discretization accuracy used in TRSK is related to the alignment index.

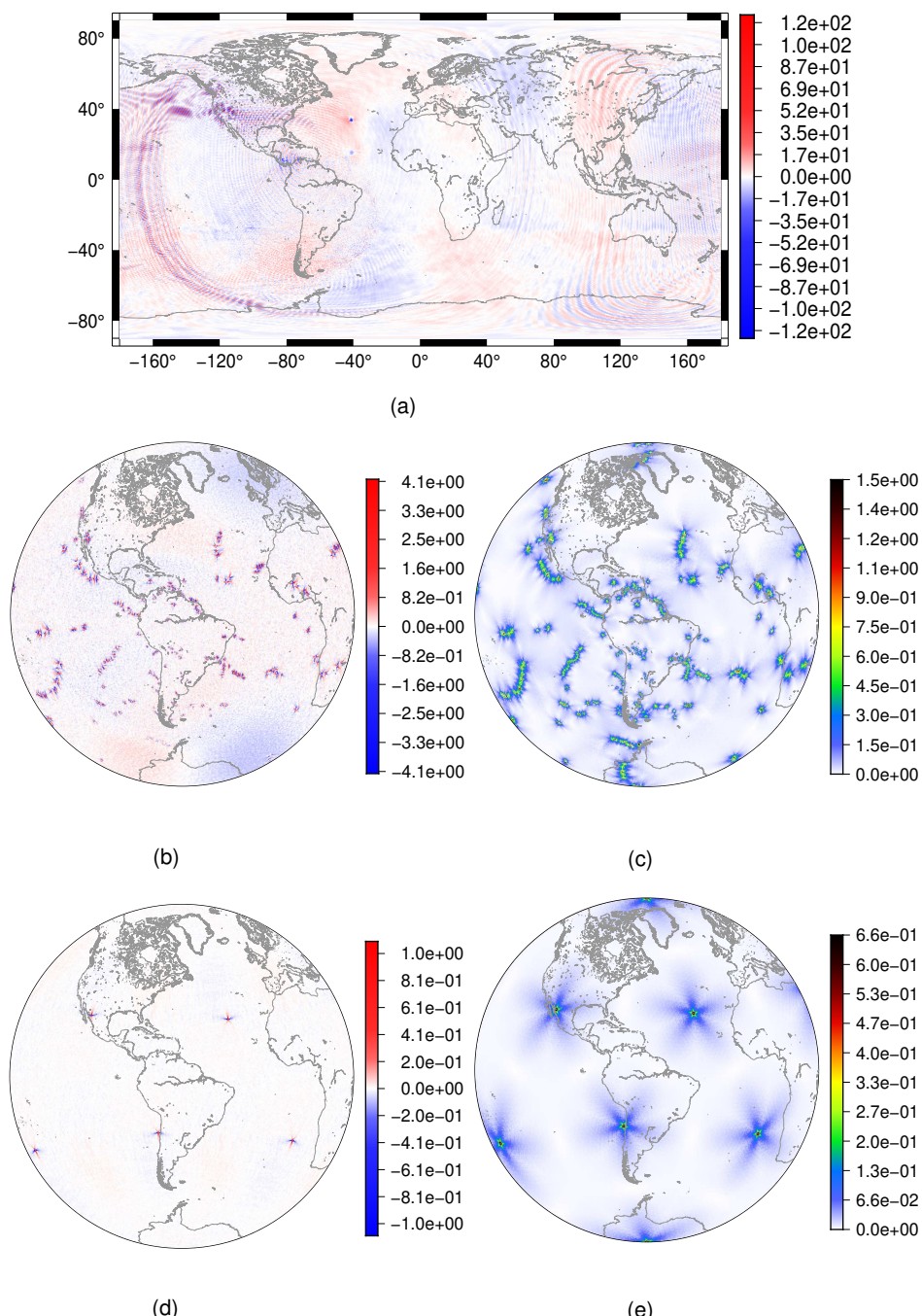

**Figure 9.** Steady geostrophic flow test case - Plots for the VR7 grid: (a): height field error for each grid cell after 30 days (without diffusion). (b): height field error for each grid cell after 30 days using numerical fourth-order hyperdiffusion based on the alignment index in the shallow-water model. (c): grid alignment index (Peixoto and Barros, 2013). Plots for the UR7 grid: (d): height field error for each grid cell after 30 days. (e): grid alignment index.

Following these analyses, we will consider in the next sections the hyperdiffusion always based on alignment and adopt for VR7 a coefficient $K_{\max} = 10^{12} m^4 s^{-1}$. For VR6 we will adopt $K_{\max} = 10^{13} m^4 s^{-1}$ and for all coarser grids (VR1 to VR5), $K_{\max} = 5 \times 10^{13} m^4 s^{-1}$ was enough to stabilize the model.

In this test case, the total energy is conserved with precision $10^{-10}$ for the UR7 grid and $10^{-9}$ for the VR7 grid without fourth-order hyperdiffusion and with precision $10^{-8}$ considering fourth-order hyperdiffusion on the VR7 grid. Overall, we can see that the addition of fourth-order hyperdiffusion did not deteriorate the total energy conservation of TRSK, since this dissipation method is a scale selective damping mechanism (Jablonowski and Williamson, 2011). We point out that the second-order diffusion $\nabla^2$ is also able to stabilize the errors in this test case. However, the operator $\nabla^2$ is not as selective as $\nabla^4$ and leads to deterioration of total energy conservation. Indeed, we re-run this test for the VR7 grid using $\nabla^2$ with constant diffusion coefficient equal to $8.2 \times 10^3 m^2 s^{-1}$, as in spectral methods with truncation level T85 (Jablonowski and Williamson, 2011), and we observe that the errors remained stable and the total energy was conserved with precision $10^{-4}$ (not shown).

Furthermore, Fig. 10 shows that adding fourth-order hyperdiffusion in the variable resolution SCVT grid simulations is needed to achieve better convergence in both $L_\infty$ and $L_2$ norms. We can see that the error does not converge to zero for both height and velocity fields, even when we add the fourth-order hyperdiffusion. Nevertheless, lack of convergence is also observed in the uniform resolution SCVT grids, although in the maximum norm, as it is shown in Peixoto (2016). We point out that Fig. 10 may suggest that the grids VR1,$\cdots$, VR5 do not develop spurious gravity waves as in the grids VR6 and VR7. Nevertheless, they form such waves but for longer times of integrations.

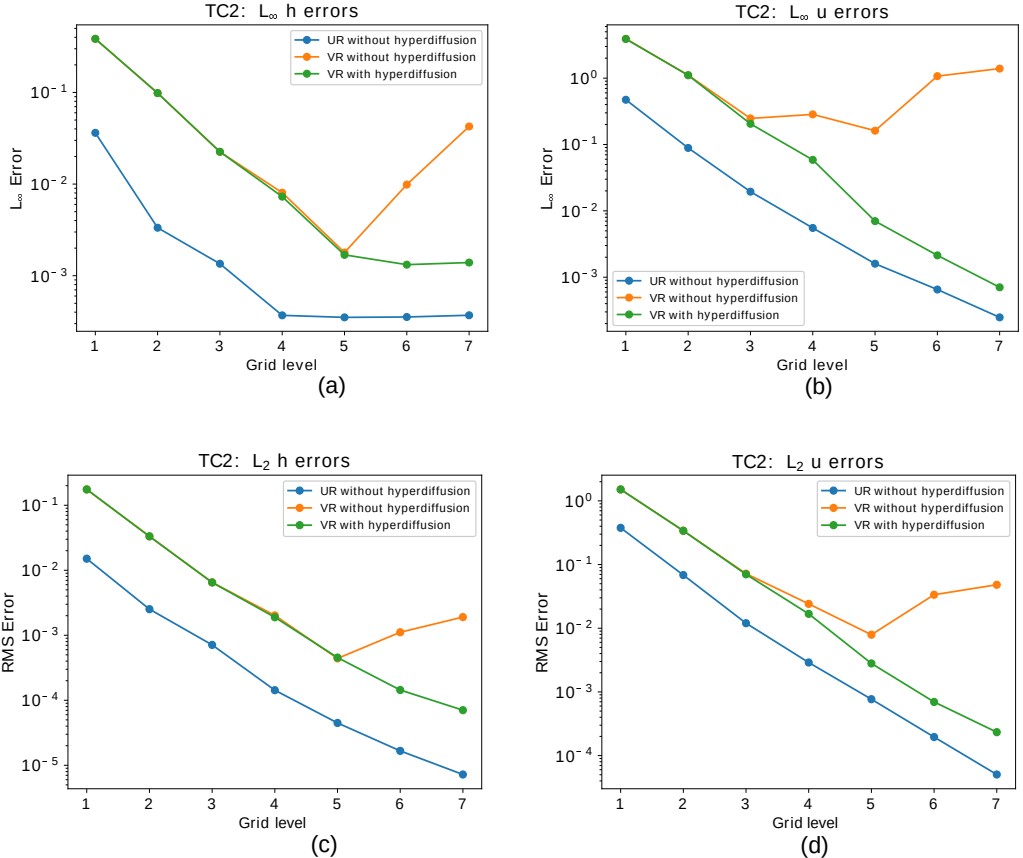

**Figure 10.** Steady geostrophic flow test case: Convergence of relative error in the $L_\infty$ norm for the height field (a) and the normal component of the velocity field (b) with respect to errors at day 30. (c) and (d) shows the same as (a) and (b), respectively, but for the $L_2$ norm. The blue lines indicate the errors for the uniform SCVT grids, while the orange lines show the error for the variable resolution grids with refinement based on topography and the green lines show the error for the same variable resolution grids but adding numerical fourth-order hyperdiffusion based on the alignment index in the shallow-water model.

In order to analyze the impact of increasing the resolution of the refined region, we consider the 1:6 ratio of fine/coarse cell diameter grid as discussed in Sect. 2.2. In Fig. 11a we illustrate the diameters (in km) of the 1:6 variable resolution grid with a grid level equal to 7. We ran the steady geostrophic flow for this grid. In Fig. 11b we show the time evolution of the relative error in the $L_\infty$ norm. The behavior is similar to the one depicted in Fig. 7a, however, the error starts to grow at an earlier time. Also, the error after 30 days is greater for the 1:6 variable resolution grid. Therefore, we can conclude that the error starts to develop earlier and is larger when we increase the ratio between fine/coarse cell diameters, which is in agreement with Liu and Yang (2017), where the authors investigate the impact SCVT grids with circular refinement region considering the classical shallow-water model. Their work suggests that ratios between high- and low-resolution cell diameters bigger than 1:3 produce

nonacceptable errors, recommending only 1:2 and 1:3 ratios. Finally, we observe that using the same amount of fourth-order hyperdiffusion based on the alignment index stabilizes the error, as the green line in Fig. 11b shows.

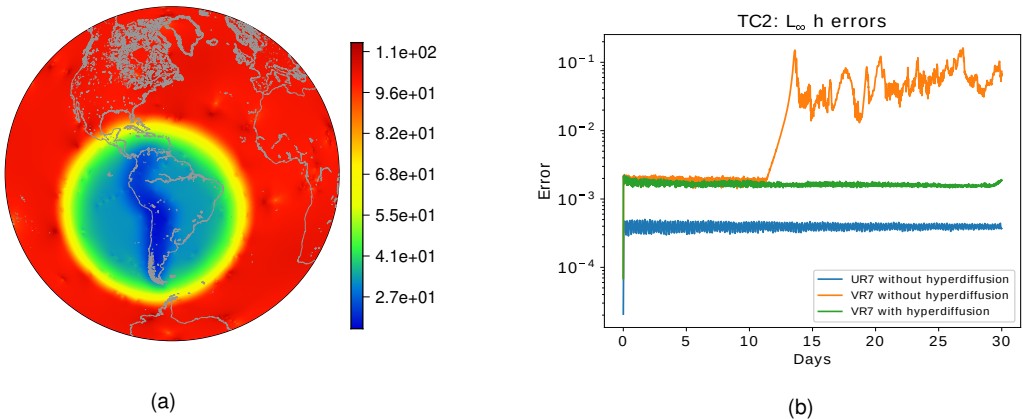

(a)

(b)

**Figure 11.** (a): diameters in km for the variable resolution grid with 1:6 coarse to fine cell ratio. The grid level is equal to 7. (b): Steady geostrophic flow test case showing the time evolution of relative error in the $L_\infty$ norm for the height field considering the variable resolution grid with 1:6 coarse to fine cell ratio. The blue lines indicate the errors for the uniform SCVT grids, while the orange lines show the error for the variable resolution grids with refinement based on topography with 1:6 coarse to fine cell ratio and the green lines show the error for the same variable resolution grids but adding numerical fourth-order hyperdiffusion based on the alignment index with $K_{\max} = 10^{12} m^4 s^{-1}$ in the shallow-water model.

#### 4.2.2 Flow over Andes mountain

This test case is an adaption of the flow over a mountain test case proposed by Williamson et al. (1992), where we replace the bottom topography $b$ with the smooth Andes topography described in Sect. 2.2. The velocity field is initialized as in the previous test case of the steady geostrophic flow. The initial height field is then given by:

$$h(\lambda, \phi, 0) = h_0 - \frac{1}{g}\left(a\Omega u_0 + \frac{u_0^2}{2}\right)\sin^2\phi - b(\lambda, \phi). \tag{24}$$

We adopted $h_0 = 5400$ and normalized the bottom topography to lie in the interval $[0, 2000]$. Since the exact solution for this test case is not available, we use as a reference solution the solution yielded by ENDGame, a semi-Lagrangian dynamical core model developed by Thuburn et al. (2010), in a latitude-longitude grid of $512 \times 1024$ points. We interpolate the values from the latitude-longitude grid to the grid points needed in SCVT grids using cubic interpolation.

We ran this test case for 30 days on the uniform and variable resolution SCVT grids of level 7 (UR7 and VR7, respectively). In Fig. 12, we show the error of the height field considering the ENDGame solution as a reference. We can notice that the error is initially concentrated on the Andes mountain (Fig. 12a) and it is propagated to the other regions of the grid (Fig. 12b-c) and finally on the whole grid (Fig. 12 (d)). A similar result is obtained in Tian (2020), where the author considers this test case using a cosine bell topography instead of the smoothed Andes topography and it is shown that this test case generates errors that propagate as gravity waves and these errors can be larger in the high-resolution grid region.

Our simulation shows that, even though the Andes is represented more accurately in the VR7 grid, the height field errors are larger in the Andes mountain in the first days of integration. Despite that, the magnitude of the error does not increase as in the steady geostrophic flow test case. However, Fig. 13a illustrates how the $L_\infty$ error for the velocity field increases with time. Even though they increase, they don't trigger larger errors as in the steady geostrophic flow test case. Figure 13b shows that the normal component of the velocity error is greater in the Africa continent, which is a coarse grid region. This velocity field error in the VR7 grid is mitigated when we add fourth-order hyperdiffusion based on the alignment index, as in Sect 4.2.1, in the shallow-water model, as Fig. 13a is showing. Again, the addition of fourth-order hyperdiffusion allowed us to have a similar temporal error evolution in both variable and uniform resolution SCVT grids, breaking the total energy conservation which is conserved with precision $10^{-10}$ without fourth-order hyperdiffusion and $10^{-8}$ with fourth-order hyperdiffusion.

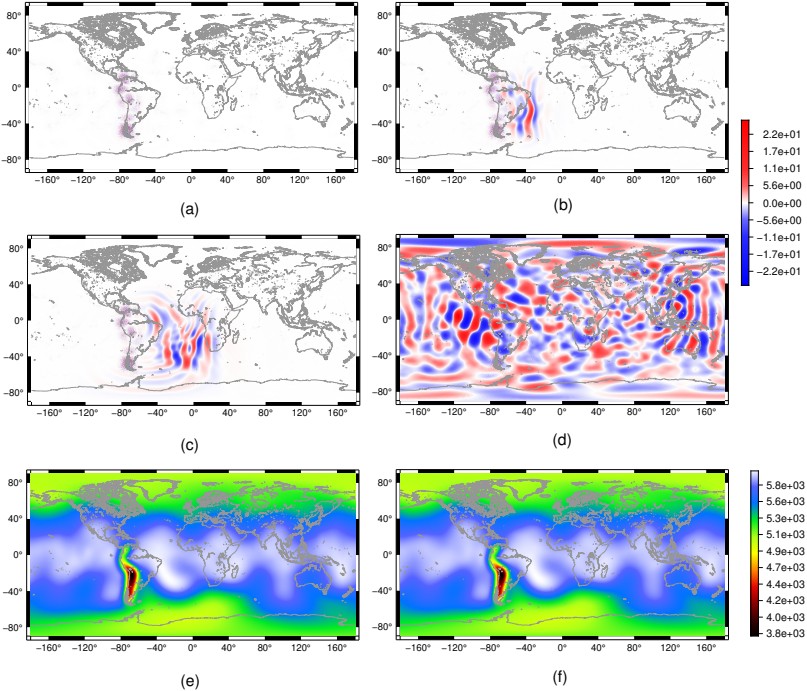

**Figure 12.** Flow over Andes mountain test case for the shallow-water model: Height field cellwise error plots for the VR7 grid after 1 hour (a), 2 days (b), 3 days (c), and 30 days (d). We are considering the ENDGame solution as a reference solution. We show the height field for the VR7 grid in (e) and the reference height field (f) after 30 days.

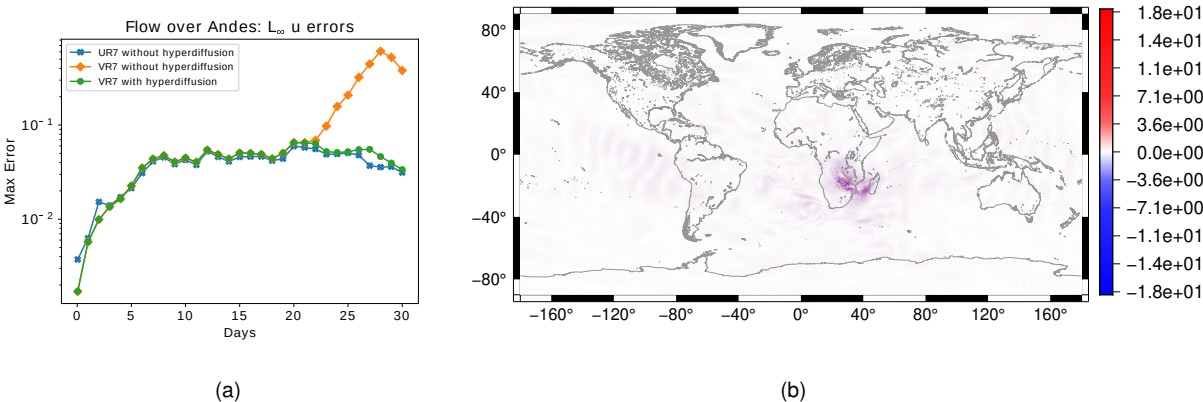

(a)                                                                                            (b)

**Figure 13.** Flow over Andes mountain test case for the shallow-water model: (a) Time evolution of relative error in $L_\infty$ norm for the normal component of the velocity field for the UR7 grid (blue line), for the VR7 with (green line) and without (orange line) numerical fourth-order hyperdiffusion based on the alignment index. (b) Cellwise error for the normal component of the velocity field at day 30 for the VR7 grid without fourth-order hyperdiffusion. We are considering the ENDGame solution as a reference solution.

### 4.2.3   Barotropic unstable zonal jet

Now, we shall analyze the test case proposed by Galewsky et al. (2004). The initial fields are given by:

$$
u(\lambda,\phi,0) = \begin{cases} 0 & \text{if } \phi \le \phi_0 \\ \frac{u_{\max}}{e_n} \exp \frac{1}{(\phi-\phi_0)(\phi-\phi_1)} & \text{if } \phi_0 < \phi \le \phi_1, \\ 0 & \phi \ge \phi_1, \end{cases}
\tag{25}
$$

$$
v(\lambda,\phi,0) = 0,
\tag{26}
$$

$$
h(\lambda,\phi,0) = h_0 - \frac{1}{g}\int^{\phi} au(\lambda,\phi',0)\left(f + \frac{\tan\phi'}{a}u(\lambda,\phi',0)\right)d\phi',
\tag{27}
$$

where we define $\phi_0 = -5°$, $\phi_1 = -45°$, $u_{\max} = 80$ and $e_n = \exp-\frac{4}{(\phi_0-\phi_1)^2}$. These parameters define the jet in the Southern hemisphere. At last, we add the following perturbation in the height field in order to trigger the instability.

$$
\hat{h}(\lambda,\phi) = \hat{h}\cos(\phi)e^{-\left(\frac{\lambda}{\alpha}\right)^2}e^{-\left(\frac{\phi-\phi_2}{\beta}\right)^2}
\tag{28}
$$

where $\hat{h} = 120$, $\alpha = \frac{1}{3}$, $\beta = \frac{1}{15}$ and $\phi_2 = -25°$. The bottom topography is set to zero.

We ran this test for 7 days for the VR7 grid and the UR7 and UR8 grids. In Fig. 14 we illustrate the potential vorticity. On day 6, in the VR7 grid, numerical noise is generated in a few grid cells on the topography-based refined region, as it is shown in Fig. 14a. This noise propagates to other grid cells at day 7 (Fig. 14b) and its magnitude reduces. These results show that the performance of the VR7 grid is problematic for this test case. For this reason, we ran again this test employing the fourth-order hyperdiffusion based on the alignment index, as in Sect 4.2.1, in the shallow-water model. As Fig. 14c shows, the numerical

noise in the VR7 grid is mitigated when we add numerical hyperdiffusion. In Fig. 14d and Fig. 14e, we depict the potential vorticity at day 7 for UR7 and UR8 grids, respectively, where no numerical fourth-order hyperdiffusion was employed. When we compare the potential vorticity at day 7 for the grids VR7 and UR7, both of level 7, with the UR8 grid, we can observe that the potential vorticity in the VR7 grid (Fig. 14c), considering numerical hyperdiffusion, represents better the vortex formed

near to Andes region than in the uniform resolution grid with the same number of cells (Fig. 14d), i.e., the vortex formed near to the Andes region in the VR7 grid with numerical fourth-order hyperdiffusion is more similar to the vortex formed in the same area in the UR8 grid (Fig. 14e) compared to the vortex generated in the UR7 grid. This simulation shows that the variable resolution grid has a good performance in this test case when we add numerical hyperdiffusion in the shallow-water model compared to the uniform grid with the same number of grid cells, showing a computational advantage of the variable

resolution grid. We do, however, highlight that since the test case is dynamically unstable, exact agreement of uniform and variable resolution grid solutions are not necessarily expected, but one may notice the similar qualitative representation of the vortex formation in both solutions.

Figure 14b shows grid-scale checker-boarding pattern in the potential vorticity. TRSK is known for having grid-scale oscillations in the potential vorticity and some potential stabilization methods have been proposed in the literature to tackle this

problem, such as the anticipated potential vorticity method (APVM) (Ringler et al., 2010) and the continuous, linear-upwind stabilized transport scheme (CLUST) (Weller, 2012). We re-run the barotropic unstable jet test case for the VR7 grid using APVM and CLUST, however, our results show that these methods do not help to mitigate the potential vorticity grid-scale oscillations.

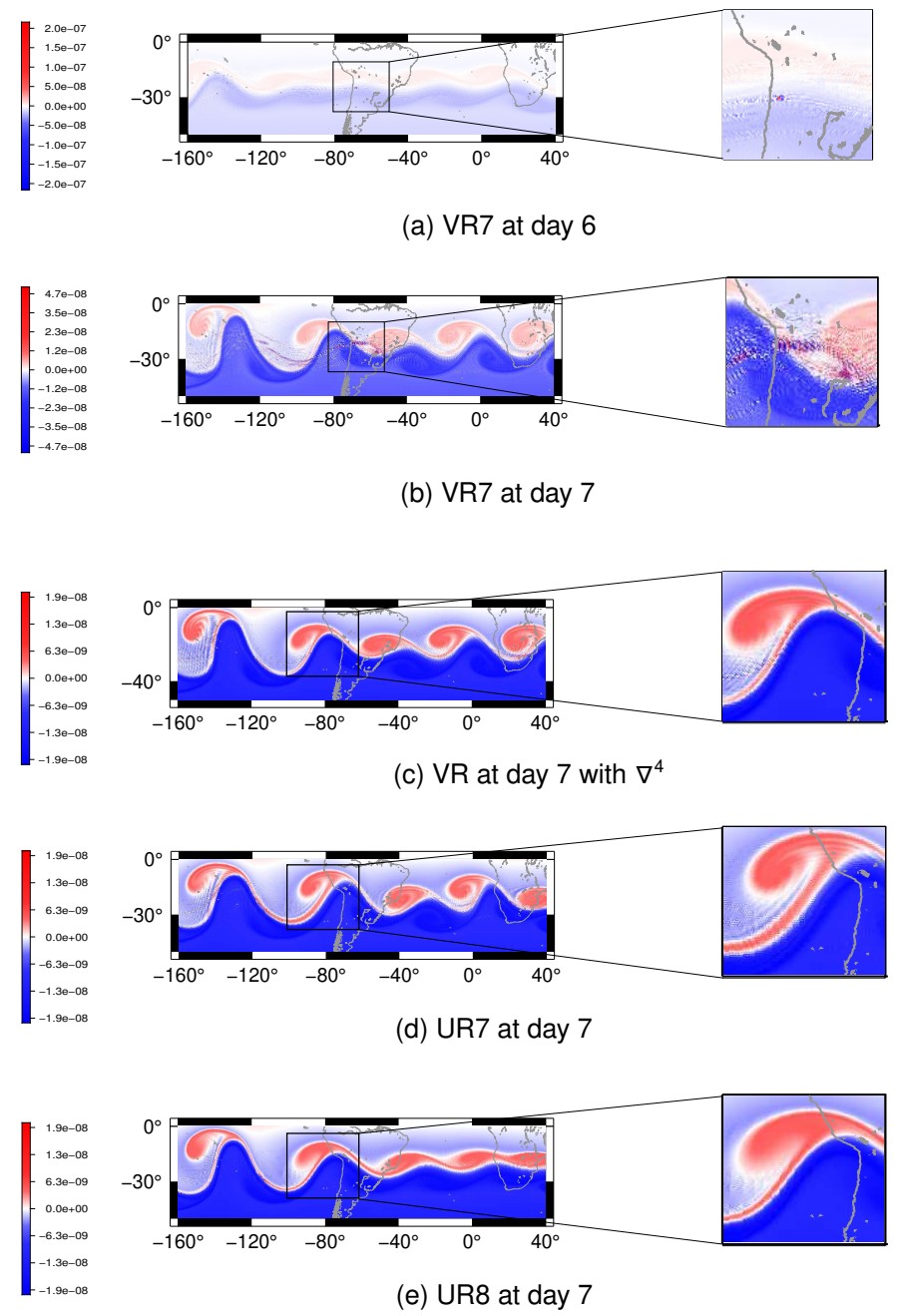

**Figure 14.** Barotropic unstable zonal jet for the shallow-water model: Potential vorticity for the VR7 grid at (a) day 6 and (b) day 7 without fourth-order hyperdiffusion. (c) shows the potential vorticity at day 7 with fourth-order hyperdiffusion in the shallow-water model for the VR7 grid. In (d) and (e) we depict the potential vorticity at day 7 for the UR7 and UR8 grids, respectively, as references.

#### 4.2.4 Matsuno baroclinic wave

This test case has been proposed by Shamir et al. (2019) and considers analytical wave solutions to the linearised shallow-water equations on the beta plane that approximate analytical solutions to the shallow-water equations on the sphere for low gravity waves speed and are taken as initial and reference solutions for this test case. It is recommended to analyse two types of waves: eastward inertia-gravity wave (EIGW, hereafter) and Rossby wave (RW, hereafter). We follow Shamir et al. (2019) analyzing both waves in this work. The mean atmosphere depth is set to $H = 30$m, the wavenumber is defined as $k = 5$ and the mode number is defined as $n = 1$. The EIGW period is $T = 1.9$ days and the RW period is $T = 18.5$ days. We recall that the time-frequency is defined by $\omega = \frac{2\pi}{T}$. For these parameters, the analytical solutions are expected to be waves that propagate in the zonal direction, where the EIGW propagates in the eastward direction and the RW propagates in the westward direction. In order to guarantee that waves pass through the whole Andes topography based locally refined region, we applied a three-dimensional rotation of $21°$ in the $x - z$ plane in the VR7 grid. We integrate the shallow-water equations for 100 wave periods for each EIGW and RW, as it is suggested in Shamir et al. (2019), for both rotated VR7 and UR7 grids to investigate any local refinement impact on the solution.

We define the variable $h' = h - H$, in order to analyze the mean atmosphere depth perturbation and for plot reasons. In Fig. 15 we depict the $h'$ field obtained for the EIGW (Fig. 15a) and the analytical $h'$ (Fig. 15b) in the VR7 grid. The same results for the RW are shown Fig. 15c and Fig. 15d. For both EIGW and RW, the error caused is due to phase error, which is a common error for many numerical schemes and similar results are obtained for the UR7 grid. Despite the waves propagating along the locally refined region, the obtained solutions preserved the wave shape even for a long time of integration.

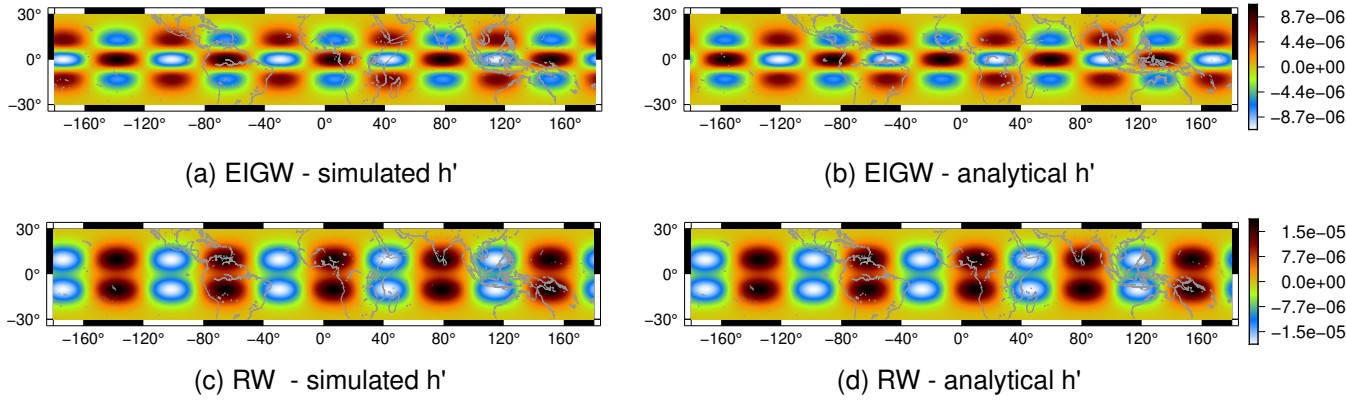

**Figure 15.** Matsuno baroclinic wave test case: Simulated $h'$ for the EIGW in the rotated VR7 grid (a) and the analytical $h'$ (b) after 100 EIGW periods (190 days). In (c) and (d) we depict the simulated and analytical $h'$ for the Rossby wave in the rotated VR7 grid after 100 RW periods (1850 days).

We also depict the Hovmöller diagrams for $h'$ considering the EIGW and RW in Fig. 16 and Fig. 17, respectively, for both rotated VR7 and UR7 grids. The latitude-time Hovmöller diagram is obtained intersecting $h'$ at $\lambda = -65°$ and the longitude-time Hovmöller diagram is obtained intersecting $h'$ at $\phi = 0°$. As Fig. 16a and (b) show, the VR7 and UR7 grids have the same

phase error for $h'$ in the EIGW test case. From Fig. 16c-d, comparing the black lines with the white line, we can conclude that
the obtained EIGW is propagating in agreement with the analytical speed for both grids since the analytical solutions propagate
zonally, the longitude-time Hovmöller diagrams are expected to be straight lines with slope $\frac{k}{\omega}$ (Shamir et al., 2019). The same
analysis holds for the RW Hovmöller diagrams depicted in Fig. 17. Therefore, the local refinement introduces no impact in the
obtained solution for this test case as we observed in the previous test cases.

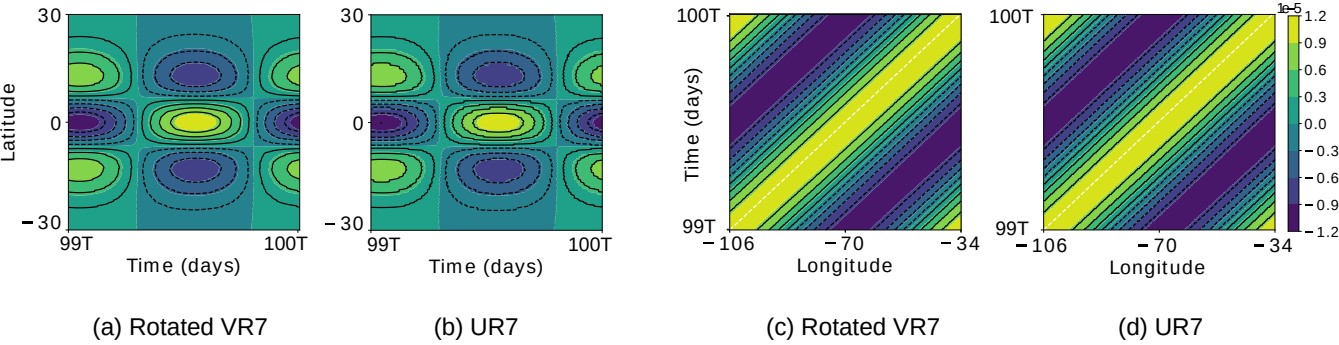

| (a) Rotated VR7 | (b) UR7 | (c) Rotated VR7 | (d) UR7 |

**Figure 16.** EIGW latitude-time Hovmöller diagrams for the simulated $h'$ in the rotated VR7 (a) and UR7 (b) grids, obtained fixing the
longitude at $\lambda = -65°$, and EIGW longitude-time Hovmöller diagrams for the simulated $h'$ in the rotated VR7 (c) and UR7 (d) grids
obtained fixing the latitude at $\phi = 0°$. $T$ denotes the period of the EIGW (1.9 days). The white dashed line has slope $\frac{k}{\omega}$, where $\omega$ and $k$ are
the EIGW time-frequency and wavenumber, respectively.

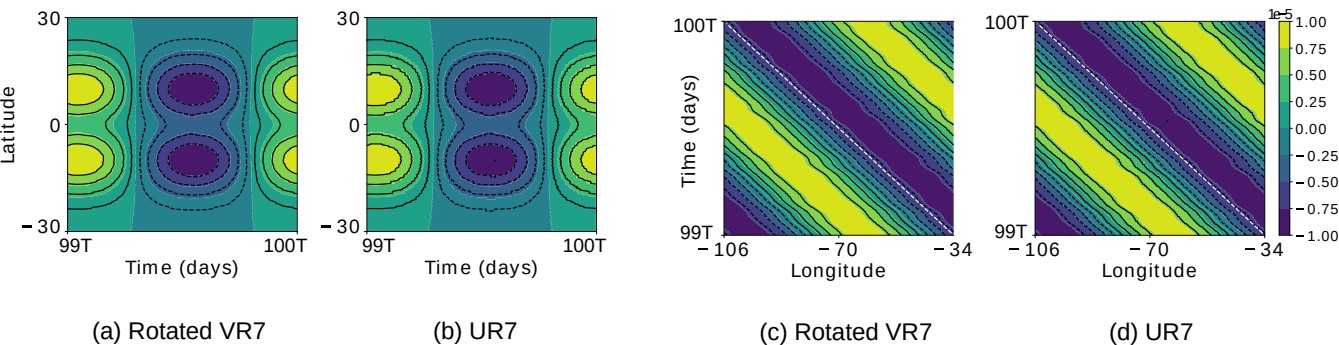

| (a) Rotated VR7 | (b) UR7 | (c) Rotated VR7 | (d) UR7 |

**Figure 17.** Same as Fig. 16 for the RW.

We remark that these test cases proposed by Shamir et al. (2019), while providing valuable insight in wave propagation
properties, seem to be little sensitive to different numerical methods. We expect that most numerical methods would perform
well on these tests. This was the case for our experiments on both uniform and locally refined grids and was also noticed in
Brachet and Croisille (2021) for a cubed-sphere based model.

### 4.3 Moist shallow-water model

We now focus our analyses of the variable resolution grids on the moist shallow-water model framework (Zerroukat and Allen, 2015), described in Sect. 3.2. We consider two test cases. The first test case is suggested by Zerroukat and Allen (2015) and is similar to flow over a mountain test case (Williamson et al., 1992). The second test case is similar to the barotropic unstable jet test case (Galewsky et al., 2004). This test case was analyzed in Ferguson et al. (2019), however, in their work, the rain formed is considered as precipitated and is removed from the model. In our analysis, we consider the rain to be advected as a tracer following Zerroukat and Allen (2015). In this section, on all test cases that use variable resolution grids, we have added the alignment-based hyperdiffusion in the momentum equation (Eq. (13)) just as in Sect. 4.1, aiming to mitigate the variable resolution numerical noises observed in Sect. 4.2. We considered the maximum value of hyperdiffusion as described in Sect. 4.2.1. At last, as we pointed out in Sect. 3.2, we discretize the tracer equations using the TRSK divergence operator, which is a centered scheme. Such centered schemes are known for having inherent grid-scale oscillations that are not necessarily removed using the dissipation mechanism proposed in this work. Despite that, this analysis will allow us to understand how the discrete divergence is affected by grid distortions.

The vapor, cloud, and rain variables in the moist shallow-water model must assume only non-negative values. For this purpose, we employed a monotonic filter to ensure that the moisture variables assume only non-negative values. At each time step of the model integration, we apply the monotonic filter for each tracer (vapor, cloud and rain), which is implemented as an iterative process. At each iteration of the filter, the tracer mass is computed for each grid cell. The grid cells with negative tracer mass are set to zero mass and its negative mass is distributed equally in the neighbor's cells. This iterative process is repeated until the maximum magnitude of the negative values is small enough. After that, all the remaining mass negative values are set to zero and the tracer value for a cell is computed as the value of its mass divided by its grid cell area. This process, even though does not conserve the tracer mass, showed to avoid negative mass and the total mass of $h(q_v + q_c + q_r)$ relative variation is less than $10^{-5}$ for the simulations that we shall see in the remaining of this section.

#### 4.3.1 Flow over a mountain

This test case is proposed in Zerroukat and Allen (2015). We only slightly changed the initial conditions by translating the initial fields in 30 degrees in the longitude in order to ensure cloud and rain formation in the South American continent. The initial height and velocity are defined as in the flow over a mountain test case for the classical shallow-water model described in Sect. 4.2.2. The initial temperature and vapor, respectively, are given by:

$$\theta(\lambda, \phi, 0) = F(\theta^{SP}, (1 - \mu_1)\theta^{EQ}, \theta^{NP}, \phi) + \mu_1 \theta^{EQ} \cos\phi \sin\left(\lambda + \frac{\pi}{6}\right), \tag{29}$$

$$q_v(\lambda, \phi, 0) = \mu_2 q_{sat}(b, h(\lambda, \phi, 0), \theta(\lambda, \phi, 0)), \tag{30}$$

where $F$ is the function given by:

$$F(f_1, f_2, f_3, \phi) = \frac{2}{\pi^2}\left[\phi\left(\phi - \frac{\pi}{2}\right)f_1 - 2\phi\left(\phi + \frac{\pi}{2}\right)\left(\phi - \frac{\pi}{2}\right)f_2 + \phi\left(\phi + \frac{\pi}{2}\right)f_3\right]. \tag{31}$$

The parameters are set as $\theta^{NP} = -40\varepsilon$, $\theta^{EQ} = 40\varepsilon$, $\theta^{SP} = -20\varepsilon$, $\varepsilon = \frac{1}{300}$, $\mu_1 = 0.05$, $\mu_2 = 0.98$. The function $q_{sat}$ is described in Sect. 3.2, Eq. (17). The topography is given by:

$$b(\lambda, \phi) = \begin{cases} 2000\left(1 - \frac{r}{r_{\max}}\right) & \text{if } r \leq r_{\max} \\ 0 & \text{otherwise.} \end{cases} \tag{32}$$

where $r = \sqrt{(\lambda - \lambda_0)^2 + (\phi - \phi_0)^2}$, $r_{max} = \frac{\pi}{9}$, $\lambda_0 = \frac{2\pi}{3}$ and $\phi_0 = -\frac{\pi}{6}$. The initial fields are shown in Fig. 18. The initial cloud and rain are set as zero.

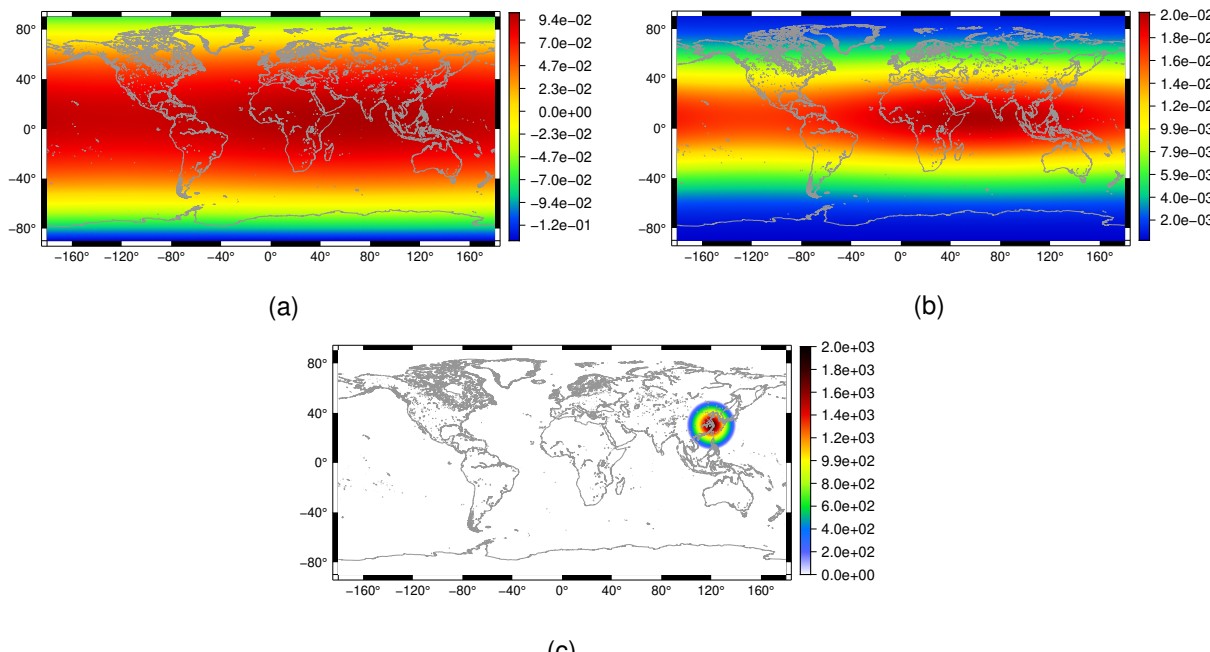

**Figure 18.** Flow over a mountain test case for the moist shallow-water model: initial conditions for the vapor (a) and temperature (b) fields, and the considered topography (c).

We ran this test for 30 days on the uniform resolution SCVT grids, using the grids UR6, $\cdots$, UR9 and the VR7 grid. In Fig. 19 we present the results for the cloud field in the South of the continent of South America at day 30 for the grids considered in our simulations. Figure 20 illustrates the same for the rain field. We employed numerical fourth-order hyperdiffusion based on the alignment index in the VR7 grid simulation as in Sect 4.2.1.

     From Fig. 19a-d and Fig. 20a-d we can notice that both cloud and rain fields are converging in the uniform resolution SCVT 515 grids. For the variable resolution SCVT grid with local refinement on the Andes mountain, the cloud field is better represented in the refined region, when we compare Fig. 19e and Fig. 19b with Fig. 19d. This illustrates a potential benefit of the locally refined grid, as the resulting clouds are well represented in the refined region while having lower computation cost with respect to the globally high-resolution uniform grid. However, when we look at the rain field in Fig. 20, we can see that the rain in

the VR7 grid (Fig. 20e) is more similar to the rain in the UR7 (Fig. 20b) than in the UR8 (Fig. 20c) or UR9 (Fig. 20d), when

we look in the topography based refined region. Notice that the formed rain in the topography-based refined region that is not formed in the UR8 and UR9 grids is present in the UR6 and UR7 grids. This indicates that the low-resolution region of the variable resolution grid is affecting the observed rain in the high-resolution region.

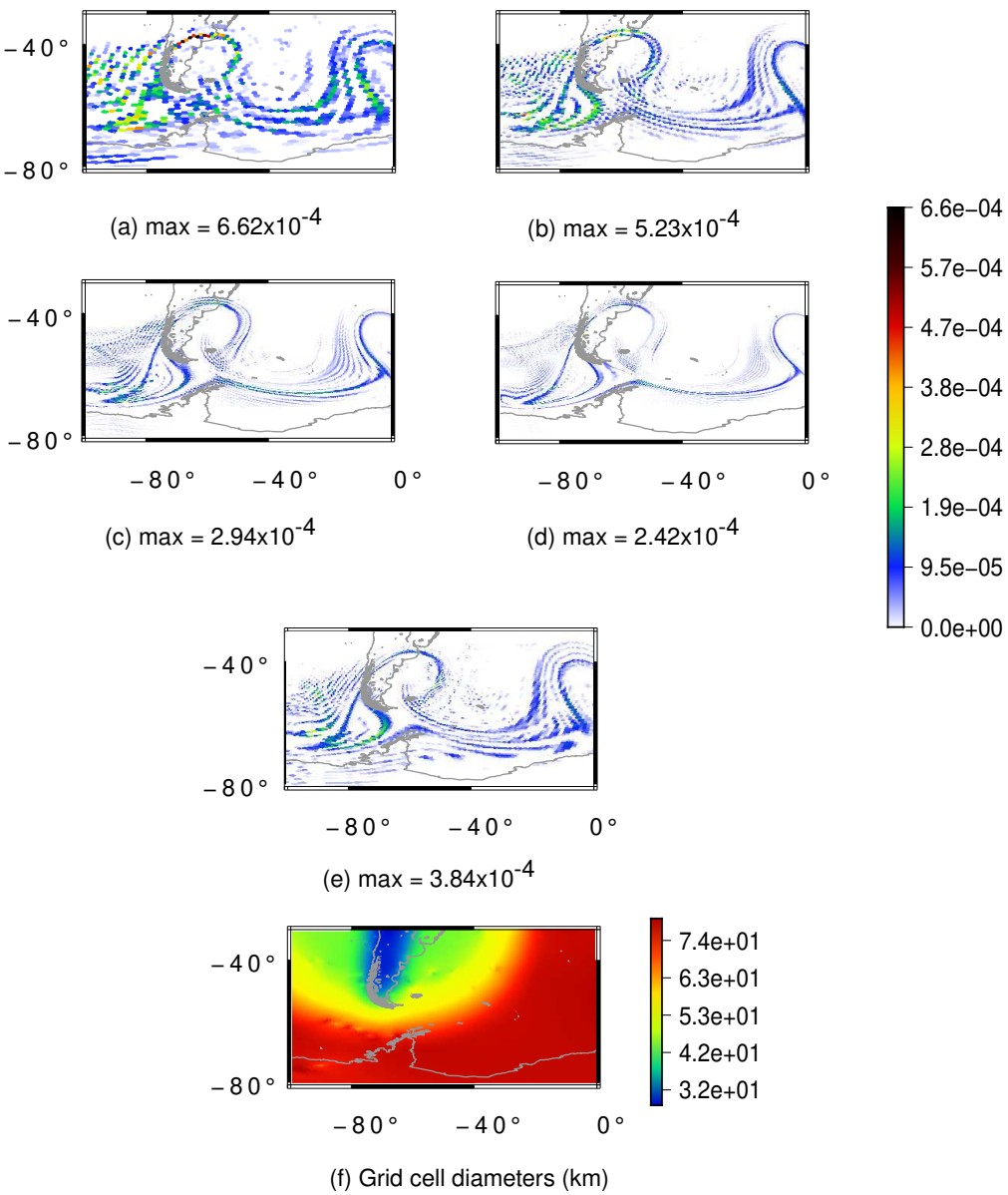

**Figure 19.** Flow over a mountain test case for the moist shallow-water model: cloud field at day 30 for the grids UR6 (a), UR7 (b), UR8 (c), UR9 (d), VR7 (e) and the cell diameters in km for the VR7 grid (f). The UR6, UR7, UR8 and UR9 grids have an average resolution of 120 km, 60 km, 30 km and 15 km, respectively.

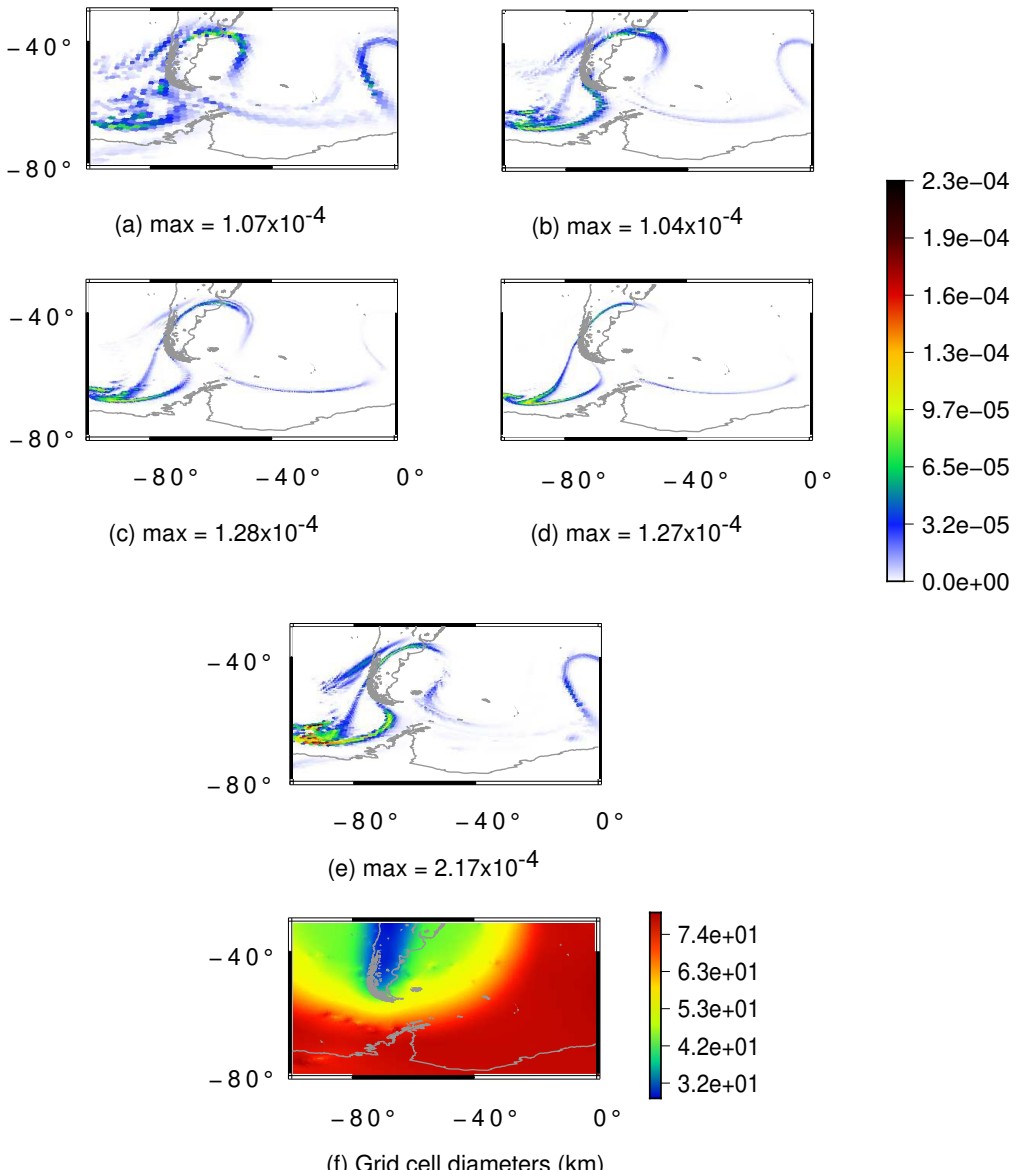

**Figure 20.** Flow over a mountain test case for the moist shallow-water model: rain field at day 30 for the grids UR6 (a), UR7 (b), UR8 (c), UR9 (d), VR7 (e) and the cell diameters in km for the VR7 grid (f). The UR6, UR7, UR8 and UR9 grids have an average resolution of 120 km, 60 km, 30 km and 15 km, respectively.

### 4.3.2 Barotropic unstable zonal jet

In this test case, we initialize the height and velocity fields as in the unstable barotropic jet for the classical shallow-water model (Sect. 4.2.3). The temperature and vapor are set as in the previous test (Eq. (29) and Eq. (30)) with $\mu_1 = 0$. This test considers the bottom topography zero on the whole sphere. A perturbation in the height field is also added and we consider the

longitude coordinated translated using the transformation $\lambda' = \lambda - 0.93\pi$ only for the perturbation since the other fields do not depend on $\lambda$.

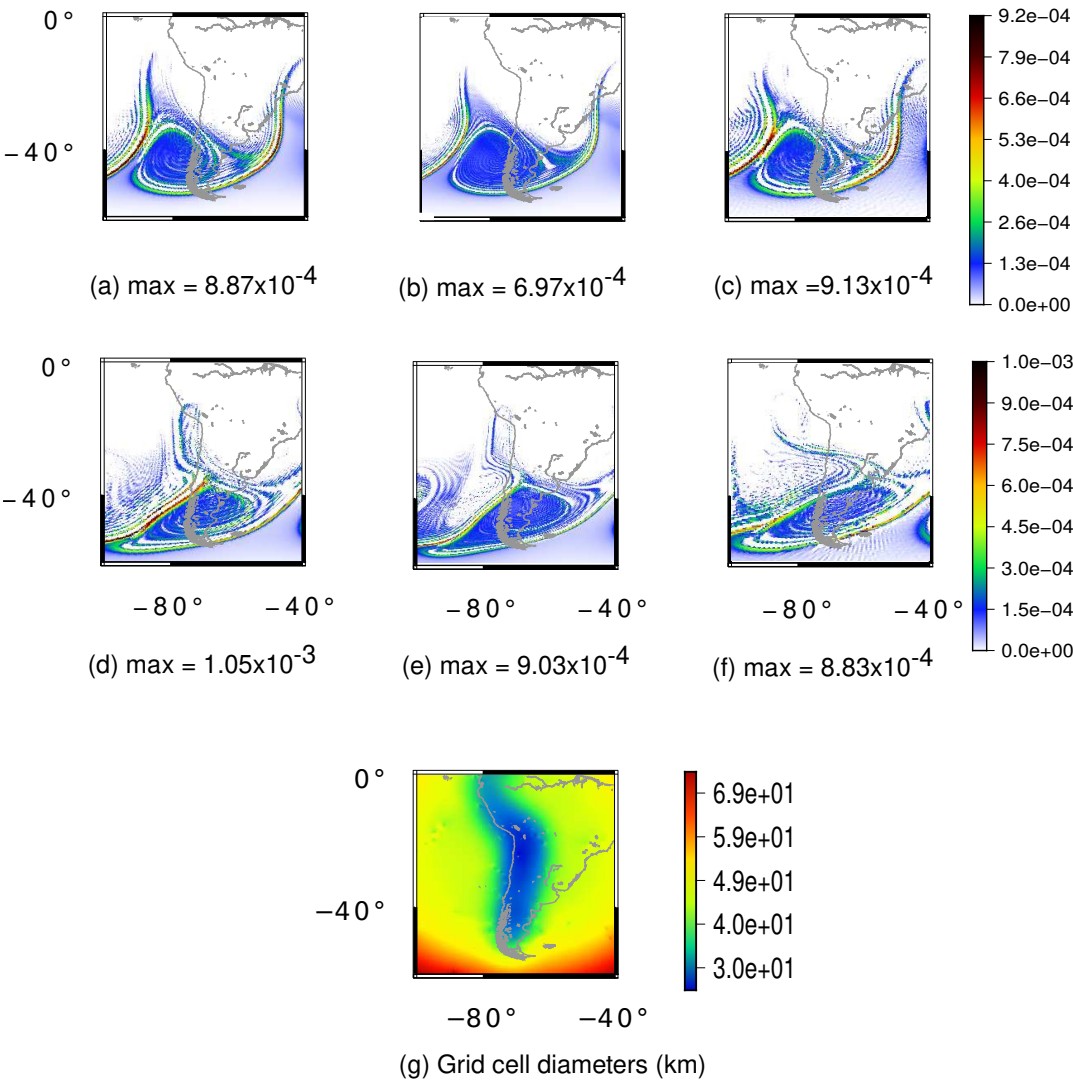

**Figure 21.** Unstable barotropic jet test case for the moist shallow-water model. (a)-(c): cloud field at day 7 for the UR8 and UR9 grids and VR7 grid, respectively. (d)-(f): cloud field at day 8 for the UR8 and UR9 grids and VR7 grid, respectively. (g): grid cell diameters in km for the VR7 grid. The UR8 and UR9 grid have an average resolution of 30 km and 15 km, respectively.

We ran this test for 8 days in the UR8, UR9 and VR7 grids, where we consider numerical fourth-order hyperdiffusion only for the VR7 grid just as the previous test case. Figures 21 and 22 show cloud and rain formation for these grids at days 7 and 8. The rain at day 7 in the variable resolution grid (Fig. 22c) has a higher magnitude when compared to the UR8 (Fig. 22a) and

UR9 grids (Fig. 22b). Nevertheless, this is expected since the region of the maximum magnitude of the rain is in the transition zone between the low and high-resolution grid. A similar analysis applies for the cloud field at day 7.

At day 8, the rain is generated on Andes mountain in the UR8 (Fig. 22d) and UR9 grids (Fig. 22e). However, this rain formation is not properly generated on the Andes region (Fig. 22f), being misplaced westward of the Andes, even though the Andes is better represented in our variable resolution grid. This illustrates how the local refinement was not able to solve the rain field in this case, and influenced the large-scale dynamics. Once again, a similar influence is observed for the cloud field on day 8.

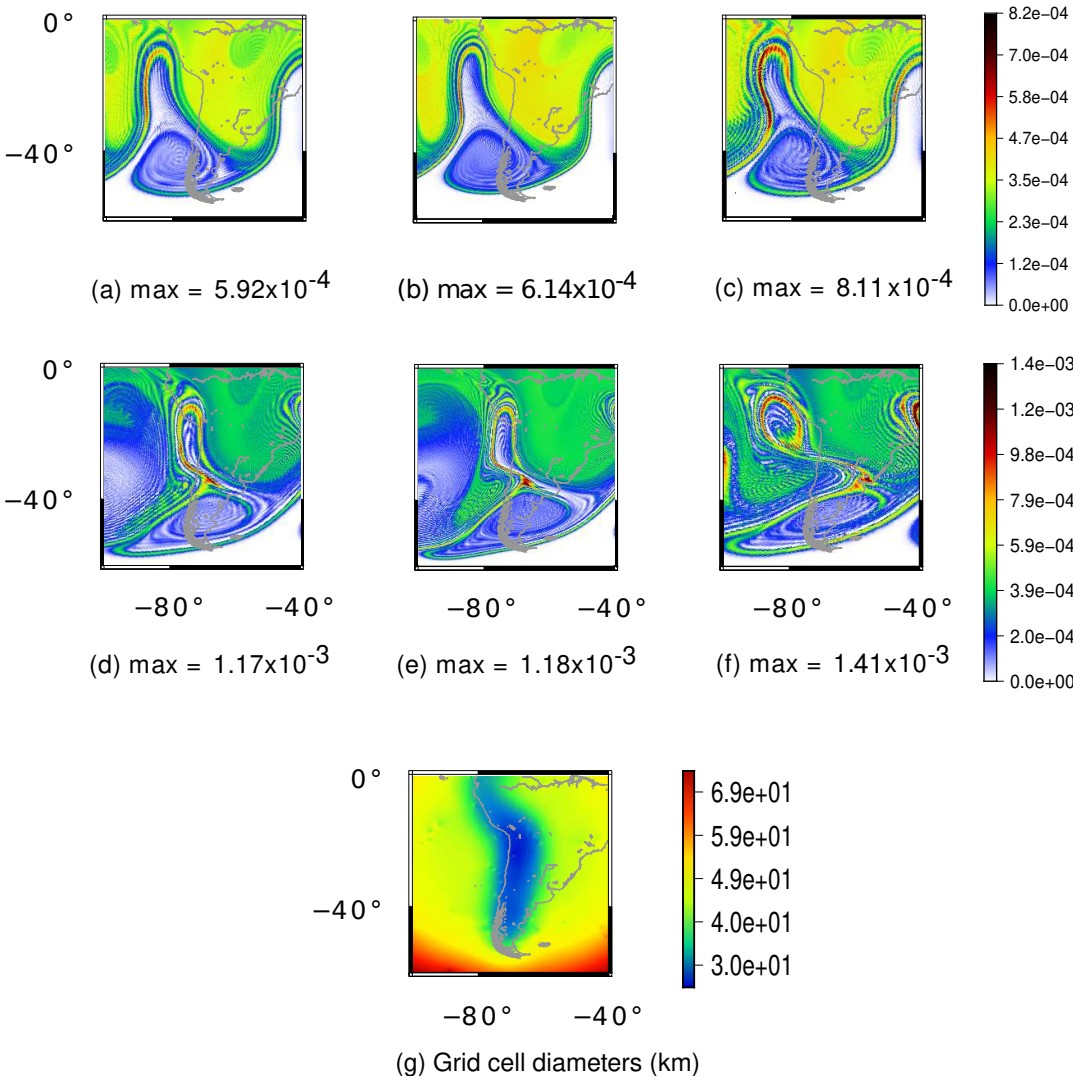

**Figure 22.** Unstable barotropic jet test case for the moist shallow-water model. (a)-(c): rain field at day 7 for the UR8 and UR9 grids and VR7 grid, respectively. (d)-(f): rain field at day 8 for the UR8 and UR9 grids and VR7 grid, respectively. (g): grid cell diameters in km for the VR7 grid. The UR8 and UR9 grid have an average resolution of 30 km and 15 km, respectively.

## 5    Conclusions

In this work, we developed SCVT grids with local refinement based on topography, that refines and captures well the Andes range and is smoothly transitioned to a circular grid region centered in South America. This circular region is again smoothly transitioned to a coarse global uniform grid. We used smoothing techniques in the Andes topography data in order to generate these grids using Lloyd's method. These grids were developed in the hope to provide a better basis for weather forecasting in South American, aiming at reduced computational cost. We evaluate the TRSK method on these grids, considering the classical shallow-water model and a moist shallow-water model proposed by Zerroukat and Allen (2015), discussing the pros and cons of the developed grids in this work.

We began our analysis with the classical shallow-water model. Although the TRSK method is designed for arbitrary orthogonal C-grids, in particular SCVT grids, we showed some of its deficiencies when considering the SCVT grid with local refinement based on topography. For instance, in test case 2 from Williamson et al. (1992), the error in the normal velocity component grows since the beginning of the model integration, triggering spurious gravity waves in the height field. This test case is known for being stable for a long time of integration in the uniform resolution SCVT grids. Therefore, our results show a negative impact on the local refinement. However, we could mitigate the error evolution of the velocity field by adding fourth-order hyperdiffusion with a coefficient based on the alignment index. The maximum of the hyperdiffusion coefficient was chosen looking for the smaller maximum coefficient that stabilized the errors considering test case 2. We also investigated constant and diameter-based hyperdiffusion coefficients and we could observe that the alignment-based has behavior similar to the constant coefficient, while adding hyperdiffusion where is needed. Therefore, we adopted the alignment-based hyperdiffusion coefficient in the further simulations. After adding hyperdiffusion, spurious gravity waves were not generated in the height field and we obtained a similar evolution of the errors with time to the ones obtained in the uniform SCVT grids. Also, we observed that when we increased the ratio between high- and low-resolution cell diameters, the errors started to develop at earlier times and the final error increased, but the same quantity of fourth-order hyperdiffusion was enough to stabilize the error. For test case 5 from Williamson et al. (1992), we replace their mountain considering a smooth Andes topography data. This test case showed that the height errors were concentrated over the Andes and generated gravity waves. This result is undesirable since the height errors were greater in the high-resolution region. We could notice that the error in the normal component of the velocity started to grow significantly after 23 days, but this was mitigated the same amount of hyperdiffusion. For the unstable barotropic jet test case from Galewsky et al. (2004), we observed that a lot of numerical noise was observed in the potential vorticity on the Andes refined region. Again, fourth-order hyperdiffusion mitigated the problem, and a vortex near to Andes formation was better represented in the variable-resolution grid in comparison with an uniform grid considering the same number of cells. From these results, we can conclude that the local refinement introduced errors in the velocity field that generate spurious gravity waves, visible in the height field. A fourth-order hyperdiffusion is necessary to be employed to stabilize the velocity field error. These results are consistent with Zhou et al. (2020), where the authors show that the employment of horizontal hyperdiffusion removes numerical noises in the transition zone of a VR SCVT grid.

The moist-shallow water model developed by Zerroukat and Allen (2015) includes moist variables in the classical shallow-water model, where conversion of vapor to cloud, cloud to vapor and cloud to rain is possible. We analyzed the TRSK method on the variable-resolution grids considering two test cases. In the first test, similar to the test case 5 from Williamson et al. (1992), we observed that the cloud field after 30 days was better represented in the variable-resolution grid of level 7 than in the uniform grid of the same level. However, for the cloud field, we notice spurious rain in the transition zone. Furthermore, spurious rain was also detectable over the Andes refined region, which we address as an interference of the coarse grid. For the unstable barotropic test case with moist, we notice that both cloud and rain fields were not properly represented in the Andes refined region after 8 days in comparison with a high-resolution solution, where we observed a misposition of the rain and cloud on the Andes region, despite the employment of numerical fourth-order hyperdiffusion.

Our work shows that, despite of its flexibility, working with general local refinements in SCVT grids using TRSK, which is also possible in the MPAS global model, requires some care. The main issues are related to spurious numerical waves that may trigger numerical instabilities, requiring numerical dissipation mechanisms to solve this problem. The requirement of dissipation is very common in most general circulation models, and TRSK has shown so far exceptionally good stability properties on quasi-uniform resolution grids. On locally refined grids, our results show that, while requiring dissipation, the imposed hyperdiffusion had very little impact on the total energy conservation of TRSK. However, the grid may influence rain and cloud formation in the refined region with the TRSK advection scheme used here for the tracers. Overall, our results indicate that locally refined grids can be effective for short periods of time (few days), but require additional caution for longer periods.

This choice of method for local grid refinement is one among several possibilities (Behrens, 2009). Nested local grid-refinement, while requiring some additional computational effort on transition zones, can preserve more regular shaped size cells, therefore less prone to such spurious modes encountered here. But it can be challenging to build a nesting grid to capture such a slim shaped featured such as the Andes Range. Static adaptative Mesh Refinement (AMR) could also be a potentially adequate solution to better represent the local feature of interest. A way to preserve the quality of cells observed in quasi-uniform grids, using static AMR to capture features of interest, is by using wavelet representation, as in Kevlahan and Dubos (2019). Overall, the most adequate way to represent the Andes Range for atmospheric modelling purposes still seems to be an open issue.

Additionally, we remark that our tracer equations were discretized here considering the same scheme as used for the continuity equation for the TRSK scheme, which is very low order. MPAS employs a high-order advection scheme for the tracer equations, which could reduce some of the issues encountered here. Results on high-order finite volume schemes, for tracer transport, on unstructured locally refined grids are to be presented in a follow-up work.

Finally, the grids developed in this work, available in the open repository https://github.com/pedrospeixoto/imodel, can be applied directly in the MPAS model, or other models that use unstructured grids, and its impact may be investigated considering the full three-dimensional model, along similar lines as Zhou et al. (2020). From one side, we expect that the unstable modes could be challenging to mitigate, due to other already existing unstable modes (e.g. Peixoto et al. (2018)). However, since such

3D models already employ a relatively large amount of diffusion or hyperdiffusion (for numerical or physical reasons), most of the issues encountered here could be adequately controlled in 3D.

*Code availability.* All codes used in this work are openly provided in the GitHub repository https://github.com/pedrospeixoto/imodel and the release version relative to results of this paper may be accessed as the Zenodo file http://doi.org/10.5281/zenodo.5579300.

*Author contributions.* PP proposed and obtained funding for the project. Both PP and LS worked on the method development and analysis. LS did the implementation of the numerical methods. Both authors wrote and revised the manuscript.

*Competing interests.* The authors declare no competing interests.

*Acknowledgements.* Financial support from São Paulo Research Foundation (FAPESP) under grant numbers 16/18445-7 and 17/25191-4 are acknowledged.

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
