# Peer review of "Topography based local spherical Voronoi grid refinement on classical and moist shallow-water finite volume models"

_Geoscientific Model Development, 2021_

## Author Comment (AC1)

**Response to Reviewers on 'Topography based local spherical Voronoi grid refinement on classical and moist shallow-water finite volume models" for GMD-2021-82**

**July 22, 2021**

We would first like to thank the referees, Darren Engwirda, Nicholas Kevlahan and an anonymous one, for the careful reading of the manuscript and constructive comments that helped to improve the paper.

In view of some of the reviewers concerns, we have reviewed our experiments and performed new simulations. The main structure of the paper was preserved, but some simulations now include different settings. We will highlight in the new version of the manuscript, in red, main changes due to reviewers comments. Small typographic changes were not highlighted in the new version to in order to focus on the more relevant changes.

In what follows we will bring in bold the original comment from the reviewers and reply in normal font below, point-by-point.

**Response to comments from Reviewer R1**

• It appears an unlimited, centred-difference scheme is employed for tracer fluxes using the 'standard TRSK' discretisation of the divergence operator when solving for advective transport. This approach can be expected to lead to significant gridscale noise and oscillation in the tracer fields, as per all results shown for the moist SWE system. As noted by the authors, the full MPAS model does not use centred-differencing for tracer transport for this reason, instead using flux-limited upwind-biased advection schemes. I feel that a better quasi-monotone advection scheme should be implemented and these runs re-analysed. With such significant grid-scale noise present, it is not clear the current comparisons between the UR and VR results for the moist SWE are necessarily as fair as they could be.

Response: We fully agree with the reviewer and acknowledge this is a limitation of the paper. Most models, including MPAS, use specially built high-order monotone tracer transport schemes, therefore the results presented here don't reflect exactly what is expected to happen in such models. However, we would like to make a few considerations on this matter. First, we have used a simple iterative positivity preserving filter in the experiments that include tracers, which already aids in some oscillation control. Naturally, this is not the best filtering for this purpose, but it is enough to reveal interesting, realistic, characteristics of the overall TRSK discretization. We are particularly interested in the behavior of the discrete divergence operator, and how variable grid distortions affect transport in this case. For instance, the same centered scheme can be successfully used in logically rectangular grids (e.g. latitudelongitude) for transport, but we see here that here the grid greatly affects the results in this case of variable resolution grids. Therefore we think that these results could be helpful in the literature.

Alongside, we have implemented several high-order monotonic finite volume tracer schemes, including new schemes, and also trying to replicate existing methods such as the one used in MPAS. However, due to the extensive discussion required to properly present the results of such finite volume schemes (see for instance Subich (2018) on effects of upwinding or centered schemes), we think it goes beyond the scope of the present paper and will be presented in a follow-up paper. Additionally, there has been some difficulty in reproducing the results of Skamarock and Gassmann (2011), from our part and also from other authors (e.g. Zhang(2018)), which is a topic currently under investigation. Therefore, bringing to this manuscript either a new method or a method that may conflict with existing literature results, would demand a major extension of the current work.

Subich, C. J. (2018). Higher-order finite volume differential operators with selective upwinding on the icosahedral spherical grid. Journal of Computational Physics, 368, 21-46.

Skamarock and Gassmann (2011). Conservative Transport Schemes for Spherical Geodesic Grids: High-Order Flux Operators for ODE-Based Time Integration, Mon. Weather. Rev., 139, 2962 - 2975.

Yi Zhang (2018). Extending high-order flux operators on spherical icosahedral grids and their applications in the framework of a shallow water model. Journal of Advances in Modeling Earth Systems, 10(1):145–164

• Is PV stabilisation (eg. APVM or LUST [3]) employed in the SWE dycore? TRSK supports an unstable-mode/grid-scale oscillation in PV dynamics that upwinded PV fluxes are needed to control. The PV contours in Fig. 10b show very significant grid-scale checker- boarding, and it would be interesting to understand the nature of PV stabilisation applied here, and to re-run with upwinding active if needed.

Thank you for pointing this out. The experiments shown in the manuscript do not include APVM or other PV stabilization methods. However, we have made experiments using both APVM and CLUST, following Weller (2012), for example in the unstable barotropic jet test case on variable resolution grids. Our results indicate that PV stabilization is not helpful to alleviate the grid-scale checker-boarding in this case. We have added a comment regarding this point on the new version of the manuscript (see section 4.1.3.).

Weller, H. (2012). Controlling the computational modes of the arbitrarily structured C grid. Monthly Weather Review, 140(10), 3220-3234.

The need to apply momentum dissipation/damping, especially for longer runs, seems consistent with practical experience. The nature/magnitude of the dissipation required is of interest though. In MPAS-O, dissipation is applied as a combination of ∇2 and ∇4 operators [2], with mesh scale dependent coefficients. Such higher-order dissipation could be expected to offer more selective damping of grid-scale noise, and potentially less impact on overall energy conservation.

Indeed the second-order Laplacian diffusion used in the first version of the manuscript can be too diffusive for the purpose. We have revisited all our experiments and indeed noticed that a fourth-order hyper-diffusion can be enough to stabilize the model, with less impact on energy conservation and error. Thank you for the suggestion. The new version of the manuscript now brings all SWE and moist SWE tests employing the  $\nabla^4$  instead of  $\nabla^2$ , and we have added a comment on previous experiments using  $\nabla^2$ . Additionally, we show below (Figure 1) the height error in the steady geostrophic flow test case after 30 days obtained considering  $\nabla^2$  and  $\nabla^4$  for different coefficients in the VR6 grid, where we can clearly see the benefits of  $\nabla^4$  in this case.

Figure 1: Height error in the steady geostrophic flow test case after 30 days with diffusion  $\nabla^2$  (left) and fourth-order hyperdiffusion  $\nabla^4$  (right) for different coefficients considering the VR6 grid.

• The barotropic jet case is known to be a challenging problem, with the nature of the turbulent vortex roll-up a strong function of mesh resolution/alignment [1,4]. I think it is interesting to use the jet as a test case, though I don't think close agreement between the UR and VR approaches should necessarily be expected.

We agree with the reviewer that the barotropic jet case is dynamically unstable, and that small changes in the early stages of the dynamics can lead to physically different dynamics later on. The VR grids may surely create enough added error to induce a physically different solution in comparison to the UR grid. However, it is important to note that the only change between the experiments is the underlying grid, therefore a numerical artifact, not a physical one. We have added a comment on the new version of the manuscript to clarify this point (see section 4.1.3)

• I feel that additional discussion/explanation on constraints on mesh generation for TRSK could be of interest, detailing, for example, why generating eg. 'wellcentred' meshes (triangles contain their own circumcentres) is important for the TRSK scheme in particular (ie. ensuring primal and dual edges intersect, 'kite' areas associated with PV remapping are positive, etc). There are many good references that could be relevant here — previous work of mine [5] is one option.

Thanks for the suggestion. We have added further information in section 2.2, better explaining the relevance of grid quality for the TRSK scheme.

**Response to comments from Reviewer R2**

• The question of how to appropriately refine computational grids is vital to avoid unacceptably large errors generated by small scale features that cannot be resolved on a uniform grid. These features may be topographical (e.g. mountains), dynamical (e.g. localized fronts or hurricanes), or related to user-defined criteria (e.g. need for more accuracy in urban areas). Topography details should be easier to deal with than dynamical features since their location is fixed and the grid refinement can be static. In this case, one may decide that the topography geometry itself is a sufficient "error indicator" to determine grid resolution. Nevertheless, the problems are not independent since topography features may generate localized waves whose influence extends beyond the topography. It is also not obvious that a grid refinement criterion based exclusively on orography is sufficient to control the error of the shallow water equations. However, this paper shows that even this relatively simply case presents challenges)

Thank you for the comments. It is indeed challenging.

• The choice of the four parameters and the sensitivity of the computations to these choices needs more justification and investigation. How sensitive are the results/errors to each of the parameters? Could their choice be optimized in some way? (Please also see questions below.)

The ratio between the diameters of cells in the high- and low-resolution regions, which in our work is given by  $\gamma$ , is one of the main factors influencing the errors. Indeed, Liu and Yang (2017) consider a circular refinement region (a similar to the region that may be obtained in our work adopting  $\lambda = 1$ ) in a SCVT grid using TRSK for the SWE, and they show that  $\gamma \geq 4$  leads to unacceptable errors, recommending only  $\gamma \leq 3$ .

We have added an experiment in section 4.1.1 with  $\gamma = 5$  in the revised paper showing that in this case, the spurious gravity waves observed in TC2 start to develop earlier and have a higher magnitude after 30 days.

The width of the transition zone  $\varepsilon$ , was also investigated by Liu and Yang (2017), and this parameter shows to have negligible impact on the errors and we expect that the same holds for our grids.

From our experience,  $\lambda = 0$ , i.e., generating a grid that refines only the Andes shape, yields larger errors for the TC2 than the one reported in the manuscript. In the manuscript, we chose the value  $\lambda$  in such a way to represent well the Andes shape and the circular refinement region in the variable resolution grid. An analysis for different values of  $\lambda \in [0, 1]$  should be definitively interesting but is out of the scope of the present study.

The parameter  $\alpha$  represents the radius of the circular refinement region and we define the value of  $\alpha$  in such a way to ensure that this region captures the South American continent. This parameter is usually constrained to the region of interest and therefore we didn't simulate our experiments for other values of  $\alpha$ .

Liu, Y. and Yang, T. (2017). Impact of Local Grid Refinements of Spherical Centroidal Voronoi Tessellations for Global Atmospheric Models, Communications in Computational Physics, 21, 1310–1324.

• This investigation considers only one example of local refinement on the Andes range (a long narrow structure with multiple steep valleys), so it is not obvious how much the results can be generalized to other cases of local refinement (e.g. around cities or topography with different gradient structures). Could you please comment on how generalizable you think the current results are?

In the present work, our main goal was not necessarily to investigate general local grid refinement and its properties. Most global models, if not all, from our knowledge, are unable to correctly capture waves structures passing through the Andes towards Brazil. This is a major practical challenge for Brazilian weather centers, that we would hope locally refined grids could help. Therefore, here we target carefully built grids aiming at representing the Andes Range and the South American continent with increased resolution. Naturally, therefore the study is not very generalizable. However, one may think that fine-scale feature-based refinements of similar granularity may suffer from similar issues as shown here.

• P 3 Nested grids and static AMR type refinement requires a bit more discussion of their advantages and disadvantages compared with the grid "stretching" approach used here. Nested grids are used extensively in numerical weather prediction, and they do not not always suffer from wave reflection at refinement boundaries. AMR techniques could also be used to provide static grid refinement and it has been used successful for dynamical adaptivity. Multi-scale or multigrid approaches, such as wavelet based adaptive refinement used in WAVETRISK, also do not show spurious effects at refinement boundaries because of the use of prolongation/restriction between scales to ensure consistency.

Thank you for pointing this out. The strategies suggested are of major interest to obtain better representations of the flow over the Andes Range. At first, nesting seems to be problematic due to the stretched nature of the Andes Range, but one could indeed develop an adequate nested grid for this profile. Here, we chose grid stretching on the SCVT form for its flexibility. Once generated, the dynamical core caries no additional cost in the numerical scheme (no need for additional interpolations, for example). Therefore, this approach seemed a natural starting point. However, we do foresee the need for investigations of more general tools, including static AMR and multi-scale (wavelet) refinements in future investigations. We have added further comments on the manuscript with respect to AMR and Nesting, but we kept it as a potential topic of future investigation.

• Is it necessary for the refined region to extend beyond the topography to ensure proper representation of dynamical features, such as waves? Because your grids are refined very smoothly, the refined regions actually do have this property (extending well beyond the Andes region). Is this necessary for dynamical accuracy as well as to guarantee good grid properties?

This is an important point that maybe was unclear in the first manuscript. We tried to clarify this in the new version (see section 2.2). Dynamical accuracy is tightly related to grid accuracy, since TRSK is very sensitive to grid irregularities. For this reason, having sharper transitions to refined regions usually led to disastrous representations of waves in our early-stage experiments. Our experienced showed that only when a smooth grid, with good quality (such as circumcenters contained within triangles, for instance), a reasonable representation of waves became evident. Therefore, we foresee the need for smooth transitions for both better grid properties and also dynamical accuracy.

• P 6 Please give more explanation and justification for the precise form of the density function in equation (7). It is clearly not the unique density function consistent with relation (4). For example, the density function depends on the topography b(x), but not (directly) on its gradient (which is presumably an important factor).

Thanks for pointing this out. The density function defined in equation (7) is consistent with equation (4) only after we set the parameters. We added a detailed discussion explaining why the density function is consistent with equation (4) in section 2.2.

For our goal at hand, considering the slim shape of the Andes, its gradient is not explicitly relevant. Since the Andes have very small plateau regions, refining with respect to the gradient would likely lead to refining everywhere within the Mountain Range. Naturally, if one were to consider refinement with respect to another mountain Range or other features of interest, gradient-based criteria could be of great importance.

• P 6 The density function (7) is based purely on the topography and will therefore not necessarily reduce errors in the solutions of the shallow water equations. However, it is also possible to optimise the choice of parameters using optimal control to minimize the height errors for certain shallow water test cases (e.g. the Williamson suite and the Galewsky jet). Have you considered a more systematic way of choosing these parameters? An optimized set of parameters may avoid the need for numerical diffusion and make better use of computational resources by refining the grid only where it is needed for accurate representation of the dynamics.

This is a very interesting point, and indeed an error-based local grid refinement could lead to better grids for the purpose at hand. At the moment, we do not have adequate mathematical and numerical tools for such kind of development. For instance, in the current development, we can anticipate that the optimal parameters for reduced error seems to lead towards the quasi-uniform grid, away from sharp gradients and localized refinement. Therefore, such development would need to carry added optimization constraints and carefully built objective functions in order to fulfill the requirement of better representation of features over the Andes.

• Figure 3 How did you choose the parameters for the refined grid shown here? Are these the choices you will use for all simulations? Given the shape of the Andes, would an elliptical (rather than circular) refinement region be a better choice? It would be helpful to include the grid refinement parameters in the caption for table 1.

The radius of the circular refinement region  $\alpha$  was defined in order to guarantee that the circular local refinement region contains the continent (South America). The ratio between low- and high- resolution  $\gamma$  and the width of the transition zone  $\varepsilon$  were defined as in Ju at el (2011). As we mentioned in the second question, larger values of  $\gamma$  lead to larger errors. Finally, the parameter  $\lambda$  was chosen empirically in such a way that the grid was able to refine both the Andes and the continent (South America).

We agree that an elliptical refinement region would interesting not solely due to the shape of the Andes but also due the shape of South American continent. However, we do not anticipate this would lead to major improvements in the model.

We have added the parameters in the caption of Table 1. These choices were used for all simulations, apart from a new test with greater difference in resolution in the refined region with respect to coarser area.

Ju, L., Ringler, T., and Gunzburger, M. (2011). Voronoi Tessellations and Their Application to Climate and Global Modeling. In: Lauritzen P., Jablonowski C., Taylor M., Nair R. (eds) Numerical Techniques for Global Atmospheric Models. Lecture Notes in Computational Science and Engineering, vol. 80, pp. 313–342, Springer, Berlin, Heidelberg.

**• P 6 What is the actual resolution in km of the ETOPO data used? How is this choice related to the resolution of the computational grids?**

The ETOPO data's actual resolution is approximately 28km for all grids. This resolution yields a good representation of the Andes shape in the density function for the grids considered here, which is the main role of the ETOPO data in this study.

• P 12 Since numerical diffusion is entirely for stabilizing purposes, why did you not use hyper diffusion, which limits diffusion to larger wavenumbers? Laplacian diffusion smooths over more wavenumbers and is often not the best choice to stabilize a computation

Thank you for pointing that out. This was really a major issue in the first manuscript. We have now adopted hyperdiffusion everywhere in the new manuscript, and discuss this choice in the main text.

• P 12 Have you tried a local diffusion coefficient that is zero outside the refined grid area?

We did not try any region specific, or scale-aware, diffusion. The main reason is that, as with grid refinement, scale-aware dissipation has all sorts of different possibilities and criteria to be chosen. This would greatly extend the paper but is surely a topic of interest. We highlight that dissipation is not necessarily tied to the grid-cell size, that is, the spurious modes do not all originate from the refined region. We know that one of the main error-prone computational cells are the ill-aligned ones, and this is a criterion that we would like to investigate in the future, including dissipation mechanisms on and around these most ill-aligned cells.

• How sensitive are the results to the precise choice of each of the grid refinement parameters? Have you done any sensitivity analysis (e.g. using Sobol' indices or Global Sensitivity Analysis GSA) on the parameters? It would be helpful to know which parameters are most influential.

This is a very important point, but very challenging. Each of these higher resolution locally refined grids can take weeks, or even months, to be generated on large computers. This is a limitation of the optimization method used in the generation of these grids, which uses a fixed point method (called Lloyd's method). Therefore, tools like GSA are almost nonviable in this case. The first locally refined grids over the Andes were generated from our group in 2017, and we have been improving grid quality based on experience over the past few years. The most relevant parameters are those related to the sharpness of changes in the grid (mainly related to  $\gamma$ ). Please see responses above with further details on parameter choices.

• Conclusions. Please comment on the extent to which the results can be generalized to a 3D hydrostatic atmosphere model, which presumably is even more sensitive to topography gradients and structure than the shallow water models considered here because of vertical fluxes.

On one side, indeed the 3D hydrostatic atmosphere model can be more sensitive to the local grid changes and topography, due to the mentioned vertical fluxes. However, most hydrostatic 3D models, such as MPAS, already naturally require added dissipation mechanisms, along with other sources of dissipation coming from physical parametrizations. Therefore, since the spurious waves encountered here are easily circumvented by small amounts of dissipation, it could be that the issues, although undesired, are not so damaging in the 3D case.

**Response to comments from Reviewer R3**

• One addition that I believe would greatly improve the paper is the inclusion of a set of grids with a greater coarse to fine cell ratio, that is, a grid generated with a greater value for  $\gamma$  in the density function (For example,  $\gamma = 6$ ). How the results would compare to those with  $\gamma = 1$  and  $\gamma = 3$ ? Are the spurious waves magnified? If so, linearly with gamma or following a higher order? Would errors start developing at earlier times? Would the inclusion of the same amount of artificial diffusion still mitigate the problem?

Thank you for pointing this out. We have generated grids with a 1:6 coarse to fine cell ratio. In Figure 2, below, we show the diameters (in km) of such a grid with a grid level equal to 7.

Figure 2: Diameters (in km) of the locally refined grid with 1:6 coarse to fine cell ratio.

As suggested by referees RC1 and RC2, we employed hyperdiffusion ( $\nabla^4$ ) instead of diffusion ( $\nabla^2$ ) as a mechanism of stabilization. We considered a constant hyperdiffusion coefficient equal to 1012. In Figure 3, below, we show the depth error evolution for the VR grids of 1:3 and 1:6 coarse to fine cell ratio with/without hyperdiffusion.

Figure 3: Steady geostrophic flow: time evolution of depth  $L_{\infty}$  error for variable resolution grids of 1:3 and 1:6 coarse to fine cell ratio with/without hyperdiffusion.

Figure 3 shows that the errors starts to develop at earlier times in the 1:6 VR grid. Also, the same amount of hyperdiffusion stabilized the error. At last, we can notice that the error

at day 30 is slightly greater in the 1:6 VR grid than in the 1:3 VR grid. We included these results in subsection 4.1.1. in the revised manuscript.

• In the steady geostrophic flow analysis, do all VR1 to VR7 results follow the same pattern of developing spurious waves? Figure 6 might suggest that is happing only for the VR6 and VR7 grids.

Yes, all the other grids from VR1 to VR7 develop the same pattern of spurious waves. The grids VR1,  $\cdots$ , VR5 form this pattern after longer times of integration, as Figure 4 below shows. Figure 6 in the original manuscript shows the error after 30 days, which explains why it suggests that this happens only for VR6 and VR7 grids.

Figure 4: Steady geostrophic flow: time evolution of depth  $L_{\infty}$  error for grids VR1, ..., VR5.

• Figure 7c shows the alignment index for the VR7 grid. What it looks like for a uniform grid, such as UR7 or UR8?

In Figure 5, below, we show the alignment index for the grids UR7 and UR8. We have added further information about this in the new version of the manuscript.

---

## Author Response (AR2)

**Letter to the Topical Editor on 'Topography based local spherical Voronoi grid refinement on classical and moist shallow-water finite volume models" for GMD-2021-82**

September 24, 2021

We would like to thank the Topical Editor for reviewing and giving constructive comments that helped to improve the paper.

The suggestion for inclusion of variable hyper-diffusion was of most relevance, and we decided to perform new analyses related to filtering requirements of the model. In turn, the new version of manuscript had all experiments re-simulated with a new diffusion mechanism. The results are qualitatevely very similar, and conclusion also, but now it is clearer that the filtering technique is as discreet as possible.

Responses to the reviewers comments were already addressed in the previous version of the manuscript, so we will highlight here only the topical editor's comments.

In what follows we will bring in blue the original comment from the topical editor and reply in normal font bellow, point-by-point.

**Response to comments from Topical Editor**

- I will mention two minor technical points: 1) In Eq. (5), phi and theta are used for latitude and longitude, while in Eq. (20), phi (different font) and lambda and used for these quantities. Since theta is used for (depth-averaged) temperature, I would use the latter convection for lat/lon throughout.

  Thank you for spotting this. We have adopted phi and lambda for the lat/lon coordinates. We also have fixed other notation mistakes in Sect. 2.2 in this review process.

- 2) My experience with hyperdiffusion is that the coefficient should depend on the local grid resolution, with more hyperdiffusion in coarse regions and less in fine regions. Please see: Guba, O., Taylor, M. A., Ullrich, P. A., Overfelt, J. R., Levy, M. N. (2014). The spectral element method (SEM) on variable-resolution grids: Evaluating grid sensitivity and resolution-aware numerical viscosity. Geoscientific Model Development, 7(6), 2803-2816.

  We agree with the Topical Editor that variable coefficient is more suitable for hyperdiffusion. The coefficient dependence on the cell size is ideal, specially from a physical point of view. Therefore, we have followed a variable fourth-order hyperdiffusion operator proposed by Klemp (2017) and we investigated not only a cell-size dependent coefficient but also a cell-geometry dependent coefficient. We performed new simulations using variable hyperdiffusion, which we give details below, and the results were included in the manuscript.

  In the revised manuscript, we have added a new section *4.1 Hyperdiffusion* where we present the variable diffusion formulation given by Klemp (2017) in Eq. 20. In this new section, we

also propose three different hyperdiffusion coefficients: constant coefficient, diameter-based and alignment-based coefficients.

The diameter-based hyperdiffusion follows Zarzycki et al. 2014, where the coefficient depends on the cell diameters and it is reduced 10 times when the cell diameters is divided by 2 (see lines 293-297 in the revised manuscript).

The alignment-based coefficient is defined using the smooth alignment index (see lines 307-313), and it was motivated by the previous versions of this work, where we could observe that the dominant factor for bigger errors is related to the cell geometry.

Both diameter- and alignment-based coefficients depend on a constant $K_{\max}$ that represents the maximum of hyperdiffusion in the coarse cells, for the diameter-based case, and in the ill aligned cells, for the alignment-based case.

We have investigated the error for the test case 2 in Sect. 4.2.1 considering different values of $K_{\max}$ and all the proposed coefficients (see Fig. 7 in the revised manuscript). We could observe that the alignment-based coefficient has a similar behavior to the constant coefficient compared to the diameter-based coefficient. Also, we observed that the error patterns (see Fig. 6) were not related to the grid resolution but to the alignment index. These experiments indicates that, even though the diameter-based hyperdiffusion may be ideal from a physical point of view, it may not be ideal for variable resolution grids. Therefore, we adopted the alignment-based hyperdiffusion for further experiments.

An optimum value of $K_{\max}$ was found for the VR7 grid (see Fig. 8) considering the alignment-based coefficient. This value of $K_{\max}$ was then adopted for all the simulations in Sect. 4.2 and 4.3. We obtained very similar results compared to the previous version of this work, where we considered constant hyperdiffusion, but now we are adding hyperdiffusion only it is necessary.

Klemp, J. B.: Damping Characteristics of Horizontal Laplacian Diffusion Filters, Monthly Weather Review, 145, 4365 – 4379, https://doi.org/10.1175/MWR-D-17-0015.1, 2017.

Zarzycki, C. M., Jablonowski, C., and Taylor, M. A.: Using Variable-Resolution Meshes to Model Tropical Cyclones in the Community 710 Atmosphere Model, Monthly Weather Review, 142, 1221 – 1239, https://doi.org/10.1175/MWR-D-13-00179.1, 2014.

- Also, note that a negative sign is necessary in the inline equation on line 276. The negative sign is necessary to ensure the hyperdiffusion operator is dissipative.

We fully agree with the Topical Editor. We have added a comment in lines 291-292 pointing out that the negative sign is necessary to ensure the dissipative behavior of the hyperdiffusion operator.

---

## Author Response (AR3)

**Letter to the referees on 'Topography based local spherical Voronoi grid refinement on classical and moist shallow-water finite volume models" for GMD-2021-82**

October 18, 2021

Once again, we would like to thank the referee, Darren Engwirda, for reviewing and giving useful comments. In what follows we will bring in blue the original comment from the referee and reply in normal font.

**Response to comments from Reviewer R2**

- The conclusion currently takes a rather negative view on variable-resolution runs, arguing that the TRSK discretisation suffers from various deficiencies that are suppressed through use of hyper-viscosity. I'd argue that, rather than being a characteristic of TRSK in particular, almost all GCMs, even those run on quasi-uniform structured grids, require added dissipation terms for numerical stability.

We agree with the referee and have adjusted the text in order to allow a fairer interpretation of the results with respect to TRSK. However, it is important to point out that TRSK was developed to be a mimetic numerical scheme, with relevant conservation properties. In principle, it should not require artificial diffusion on regular grids for stability reasons, except on baroclinic regimes (small equivalent depths). Our results show that, in shallow-water model frameworks, the variable resolution grids require the addition of numerical hyperdiffusion for numerical stability, while the quasi-uniform grids do not. We find it is important to highlight that grid flexibility comes with a price, but we have amended the text (see lines lines 584-587) to correctly highlight that the price is not very high, especially compared to many other GCM.

- The issue, in my view, is mainly related to the discretisation of the advective tendencies (thickness, PV, tracers, etc) with dispersion errors (oscillations, waves) guaranteed to be generated by Arakawa-type schemes that use centred-differences. Adding hyper-viscosity is one popular solution to this problem, with upwind/flux-limited formulations being another.

Thank you for the comment. We agree that centred-differences can be a major source of potential instabilities. In the case of TRSK, these discretizations were carefully taylored to preserve certain numerical properties of the momentum-continuity-PV equations, which are not easily achievable via non-centred approaches (see, for instance, Subich 2018). We found that adaptive (geometry dependent) hyper-viscosity seems to provide a good trade-off between stabilization and mimetic property preservation, but indeed other approaches could be taylored for the purpose.

Subich, C.J., 2018. Higher-order finite volume differential operators with selective upwinding on the icosahedral spherical grid. Journal of Computational Physics, 368, pp.21-46.

- In the moist-SWE results in 4.3, it's not clear whether numerical dissipation is added to the tracer equations, or just to the momentum balance?

Thank you for pointing this out. The numerical dissipation was added just to the momentum equation as in section 4.2, aiming to avoid the grid-related errors observed in the variable resolution grids. We have clarified this in lines 480-483.

- I don't believe it should be expected that adding $del^4(u)$ to the momentum equations will remove the grid-scale oscillations from the rain/cloud tracers, since I expect these effects will be caused by the centred tracer advection scheme used.

As we pointed out previously, the addition of $del^4(u)$ to the momentum equation aims to mitigate numerical noises triggered by the velocity field and preserve the stability of main flow equations (continuity-momentum equations). So, indeed, we do not expect that adding $del^4(u)$ will prevent grid-scale oscillations from the tracers, inherent from the centered scheme advection. We believe that, anyhow, these results reveal a relevant aspect of the discrete divergence, also used in other sub-grid scale physical processes. We have added a comment regarding this in lines 483-486.